# The anti-hypertensive drug prazosin inhibits glioblastoma growth via the PKCδ-dependent inhibition of the AKT pathway

Suzana Assad Kahn[1,2,3,4], Silvia Lima Costa[1,2,3,5], Sharareh Gholamin[4], Ryan T Nitta[4], Luiz Gustavo Dubois[1,2,3,6], Marie Fève[7], Maria Zeniou[7], Paulo Lucas Cerqueira Coelho[1,2,3,5], Elias El-Habr[1,2,3], Josette Cadusseau[8], Pascale Varlet[9,10], Siddhartha S Mitra[4], Bertrand Devaux[1,2,10,11], Marie-Claude Kilhoffer[7], Samuel H Cheshier[4], Vivaldo Moura-Neto[6], Jacques Haiech[7], Marie-Pierre Junier[1,2,3,†] & Hervé Chneiweiss[1,2,3,*,†]

## Abstract

A variety of drugs targeting monoamine receptors are routinely used in human pharmacology. We assessed the effect of these drugs on the viability of tumor-initiating cells isolated from patients with glioblastoma. Among the drugs targeting monoamine receptors, we identified prazosin, an α1- and α2B-adrenergic receptor antagonist, as the most potent inducer of patient-derived glioblastoma-initiating cell death. Prazosin triggered apoptosis of glioblastoma-initiating cells and of their differentiated progeny, inhibited glioblastoma growth in orthotopic xenografts of patient-derived glioblastoma-initiating cells, and increased survival of glioblastoma-bearing mice. We found that prazosin acted in glioblastoma-initiating cells independently from adrenergic receptors. Its off-target activity occurred via a PKCδ-dependent inhibition of the AKT pathway, which resulted in caspase-3 activation. Blockade of PKCδ activation prevented all molecular changes observed in prazosin-treated glioblastoma-initiating cells, as well as prazosin-induced apoptosis. Based on these data, we conclude that prazosin, an FDA-approved drug for the control of hypertension, inhibits glioblastoma growth through a PKCδ-dependent mechanism. These findings open up promising prospects for the use of prazosin as an adjuvant therapy for glioblastoma patients.

**Keywords** glioma; GL261; rottlerin; sh PKCδ; δV1.1
**Subject Categories** Cancer; Neuroscience

## Introduction

Glioblastoma is the most common and aggressive form of primary malignant brain tumors. Highly vascularized, infiltrating, and resistant to current therapies, glioblastomas affect patients at different ages with a median survival shorter than 18 months (Schechter, 1999).

Isolation of tumor cells with stem-like properties from glioblastoma has resulted in a novel understanding of tumor behavior. These glioblastoma-initiating cells (GICs) exhibit long-term self-renewal and initiate tumors, contributing to the generation of all subtypes of cells that compose the tumor (Chen et al, 2012b; Cheng et al, 2013). A growing body of evidence implies GICs as crucial determinants of tumor behavior, including proliferation, invasion, and—most importantly—as major culprits of glioblastoma resistance to the standard of care treatments (Bao et al, 2006; Stupp & Hegi, 2007; Murat et al, 2008; Diehn et al, 2009; Chen et al, 2012a). Thus, targeting GICs represents one of the main therapeutic challenges to significantly improve glioblastoma treatments.

Investigation of GIC properties has led to early recognition of their sensitivity to molecular changes in the micro-environment, exemplified by the loss of their stem and tumor-initiating properties

1   INSERM, UMR-S 1130, Neuroscience Paris Seine-IBPS, Paris, France
2   CNRS, UMR 8246, Neuroscience Paris Seine-IBPS, Paris, France
3   Sorbonne Universités, UPMC Université Paris 06, UMR-S 8246, Neuroscience Paris Seine-IBPS, Paris, France
4   Department of Neurosurgery, Institute for Stem Cell Biology and Regenerative Medicine and Division of Pediatric Neurosurgery, Lucile Packard Children's Hospital, Stanford University, Stanford, CA, USA
5   Neurochemistry and Cell Biology Laboratory Universidade Federal da Bahia, Salvador-Bahia, Brazil
6   Instituto Estadual do Cérebro Paulo Niemeyer, Rio de Janeiro, Brazil
7   Laboratoire d'Innovation Thérapeutique, Laboratoire d'Excellence Medalis, Faculté de Pharmacie, Université de Strasbourg/CNRS UMR7200, Illkirch, France
8   UMR INSERM 955-Team 10, Faculté des Sciences et Technologies UPEC, Créteil, France
9   Department of Neuropathology, Sainte-Anne Hospital, Paris, France
10  Paris Descartes University, Paris, France
11  Department of Neurosurgery, Sainte-Anne Hospital, Paris, France
   *Corresponding author. Tel: +33 1 44 27 52 94; E-mail: herve.chneiweiss@inserm.fr
   †These authors contributed equally to this work

following serum treatment (Singh *et al*, 2003; Lee *et al*, 2006; Gunther *et al*, 2008; Liu *et al*, 2009; Silvestre *et al*, 2011). Within the brain, tumor cells are exposed to extracellular signals from the nervous parenchyma. GICs and endothelial cells have been shown to exert reciprocal control of their properties through release of extracellular factors (Galan-Moya *et al*, 2011; Thirant *et al*, 2012). Of special interest are molecular signals arising from diffuse neuro-transmitter systems, which can affect GIC behavior. These monoaminergic modulator systems control broad central nervous system functions such as arousal, sleep, food intake, and mood, which are disrupted in several neuropathological situations. Although α1-adrenergic receptor (α1-AR) agonists stimulate rodent neural stem cell (NSC) proliferation and protect them from stress-induced death (Ohashi *et al*, 2007; Gupta *et al*, 2009), their effects on GICs are unknown.

Many molecules targeting the G protein-coupled receptors (GPCR) of these monoaminergic systems have been developed and used safely and effectively for over 40 years in human pharmacology. Determining whether GIC properties could be manipulated by such pharmacological compounds should help proposing adjuvants to current chemotherapies, or conversely identifying treatments that may promote tumor progression.

Here, we found that prazosin, a non-selective α1-AR and a selective α2B-AR antagonist, induced apoptosis in patient-derived GICs *in vitro*, and inhibited expansion of tumors initiated by GICs *in vivo*. The effect of prazosin occurred via a PKCδ-dependent inhibition of AKT pathway. This effect was independent from adrenergic receptors, revealing a novel off-target activity of prazosin and a novel therapeutic application for this FDA-approved drug.

## Results

### Prazosin induces GIC death and inhibits glioblastoma growth

In this study, we used two collections of patient-derived GICs endowed with stem-like properties isolated in two distinct laboratories (Patru *et al*, 2010; Silvestre *et al*, 2011; Fareh *et al*, 2012; Thirant *et al*, 2012). A major feature of these cells is their resistance to the currently used chemotherapy temozolomide (Patru *et al*, 2010). The effects of α-AR antagonists on GIC viability were determined following a 3-day treatment on a patient-derived GIC culture, TG1 (Patru *et al*, 2010; Fareh *et al*, 2012; Thirant *et al*, 2012). The following antagonists, all known to act as α-AR antagonists in the nanomolar range (http://www.bindingdb.org/bind/ByLigandName.jsp), were used: prazosin (α1-AR and α2B-AR antagonist), BMY 7378 (α1D-AR antagonist), terazosin (α2B-AR antagonist), ARC 239 (α2B-AR antagonists), and doxazosin (α1-AR antagonists). Only prazosin inhibited GIC viability in a robust and concentration-dependent manner (Fig 1A). Prazosin-induced GIC death was also observed after 24 h of treatment (Fig 1B). Prazosin-induced cell death was observed in all patient-derived GIC cultures tested (TG1, TG16, GBM5, and GBM44, Fig 1C). Of note, either GICs bearing the wild-type (e.g. TG1) or a mutant form of *TP53* lacking DNA binding activity (e.g. TG16) (Silvestre *et al*, 2011) were sensitive to prazosin treatment (Fig 1C), indicating that prazosin effect was independent of the transcription factor P53, a well-known regulator of cell survival. In addition, we explored whether GICs having escaped a first 72-h prazosin treatment were responsive to a second prazosin treatment. The results showed that GICs remained sensitive to 30 μM prazosin (Fig EV1D). The viability of human fetal brain-derived neural stem cells/neural progenitor cells (NSC24, NSC25, NSC5031, and NSC8853), on the other hand, was only marginally decreased at prazosin concentrations of 10 μM or higher (Fig 1C). Extreme limiting dilution assay (ELDA) was used to further evaluate the targeting of GICs by prazosin. Frequency of sphere-forming cells, a surrogate property of GICs (Flavahan *et al*, 2013), was drastically reduced by prazosin, dropping from 1/3.88 to 1/248 for TG1 ($P = 1.13 \ 10^{-10}$) and from 1/6.32 to 1/31 for GBM44 ($P = 0.0331$) (Figs 1D and EV2). In addition, we sorted the GIC according to their expression of EGFR, a marker of malignancy, and of CD133 and CD15, frequently used as GIC markers (Son *et al*, 2009). Prazosin also inhibited the survival of every population subtype, including EGFR[+]/CD133[+]/CD15[+] cells (Fig 1E). To further evaluate whether the effectiveness of prazosin is influenced by the stem and/or differentiated state of the cells, NSCs and GICs were differentiated along the astroglial, oligodendroglial, and neuronal lineages (Fig 1F). Prazosin inhibited also the survival of differentiated glioblastoma cells while minimally affecting differentiated NSCs (Fig 1G).

We then assessed the *in vivo* effect of prazosin on orthotopic glioblastoma xenografts from GICs derived from human glioblastoma samples (GBM5 and GBM44). EGFR[+]/CD133[+] cells, which

**Figure 1. Prazosin inhibits GIC survival.**

A   Viability analysis of GICs treated with the α-AR antagonists prazosin, ARC 239, doxazosin, BMY 7378, and terazosin. GICs were treated with the antagonists or corresponding vehicles for 72 h, and viability was assayed using WST-1. *$P < 0.05$, $n = 4$, two-sided Mann–Whitney *U*-test.

B   Quantification of GIC survival using trypan blue exclusion after 24 and 72 h of treatment with prazosin. *$P = 0.0286$, $n = 4$, two-sided Mann–Whitney *U*-test.

C   Viability analysis of patient-derived GICs (TG1, TG16, GBM5, GBM44) and NSCs (NSC24, NSC25, NSC5031, NSC8853) treated with prazosin for 72 h. *$P = 0.0286$, $n = 4$, two-sided Mann–Whitney *U*-test.

D   Analysis of the sphere-forming capabilities of GICs using the extreme limiting dilution assay. Cells were seeded in presence of vehicle or 30 μM prazosin (PRZ). Sphere formation was scored 10 days post-seeding. Frequency of sphere-forming cells: Control = 1/3.88 (lower 8.61, upper 1.95); prazosin 1/248 (lower 1,003, upper 62), $n = 12$, $P = 1.13 \times 10^{-10}$. Overall test for difference in stem cell number between groups.

E   Viability analysis of GICs treated with prazosin after sorting cells according to their expression of EGFR, and the neural stem cell markers CD15 and CD133. Prazosin inhibits cell viability regardless of CD133 or CD15 expression. *$P = 0.0286$, $n = 4$, two-sided Mann–Whitney *U*-test.

F   Immunocytochemical staining of NSCs and GICs cultured in media favoring neuronal (β3-tubulin), astroglial (GFAP), or oligodendroglial (O4) differentiation (Diff media). β3-tub: β3-tubulin. Scale bar: 20 μm.

G   Viability analysis of differentiated NSCs and GICs treated with prazosin for 72 h. Diff: differentiation. *$P = 0.0286$, $n = 4$, two-sided Mann–Whitney *U*-test.

Data information: Results in (A–C, E, G) are presented as mean ± SD in biological quadruplicates from three independent experiments.

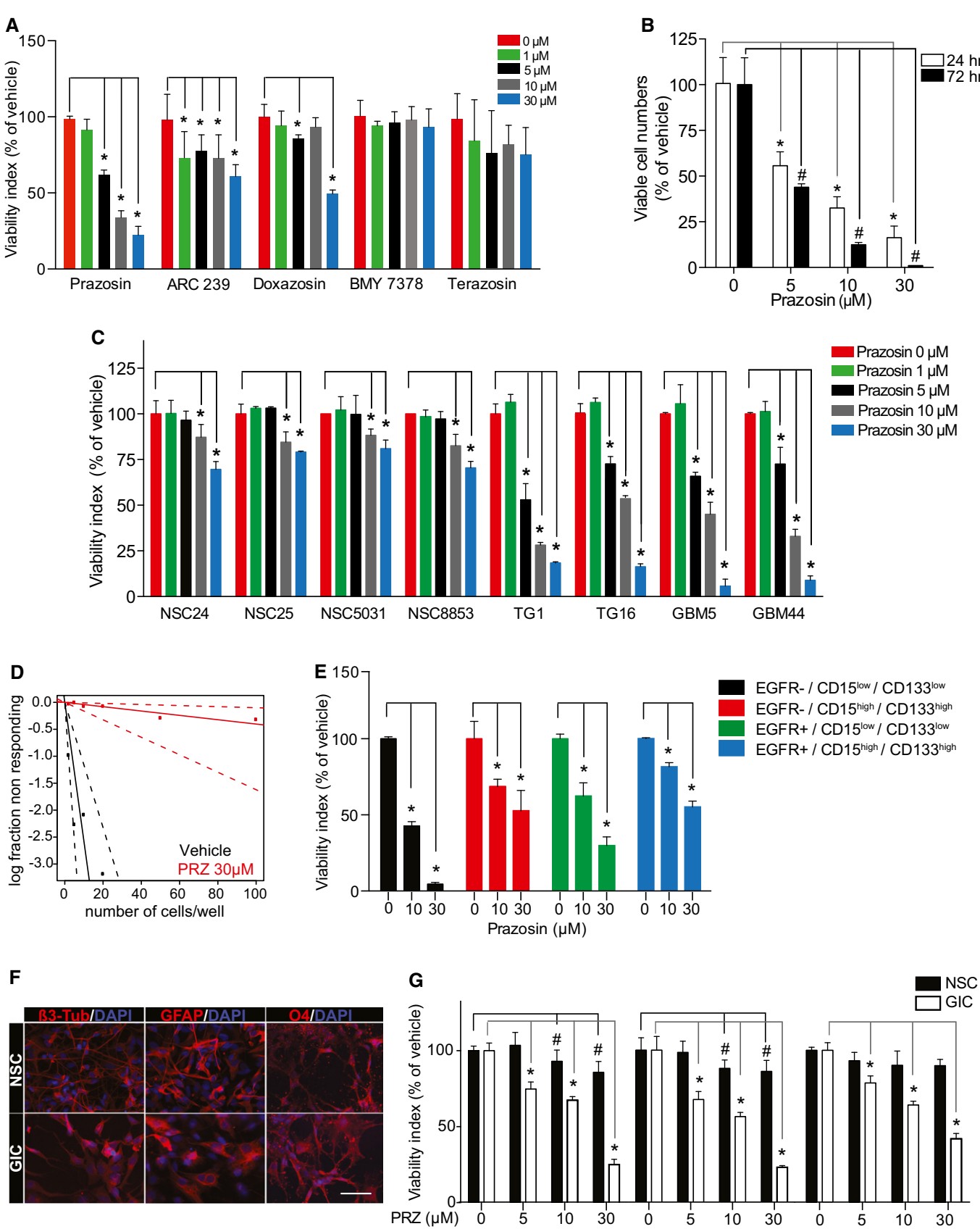

**Figure 1.**

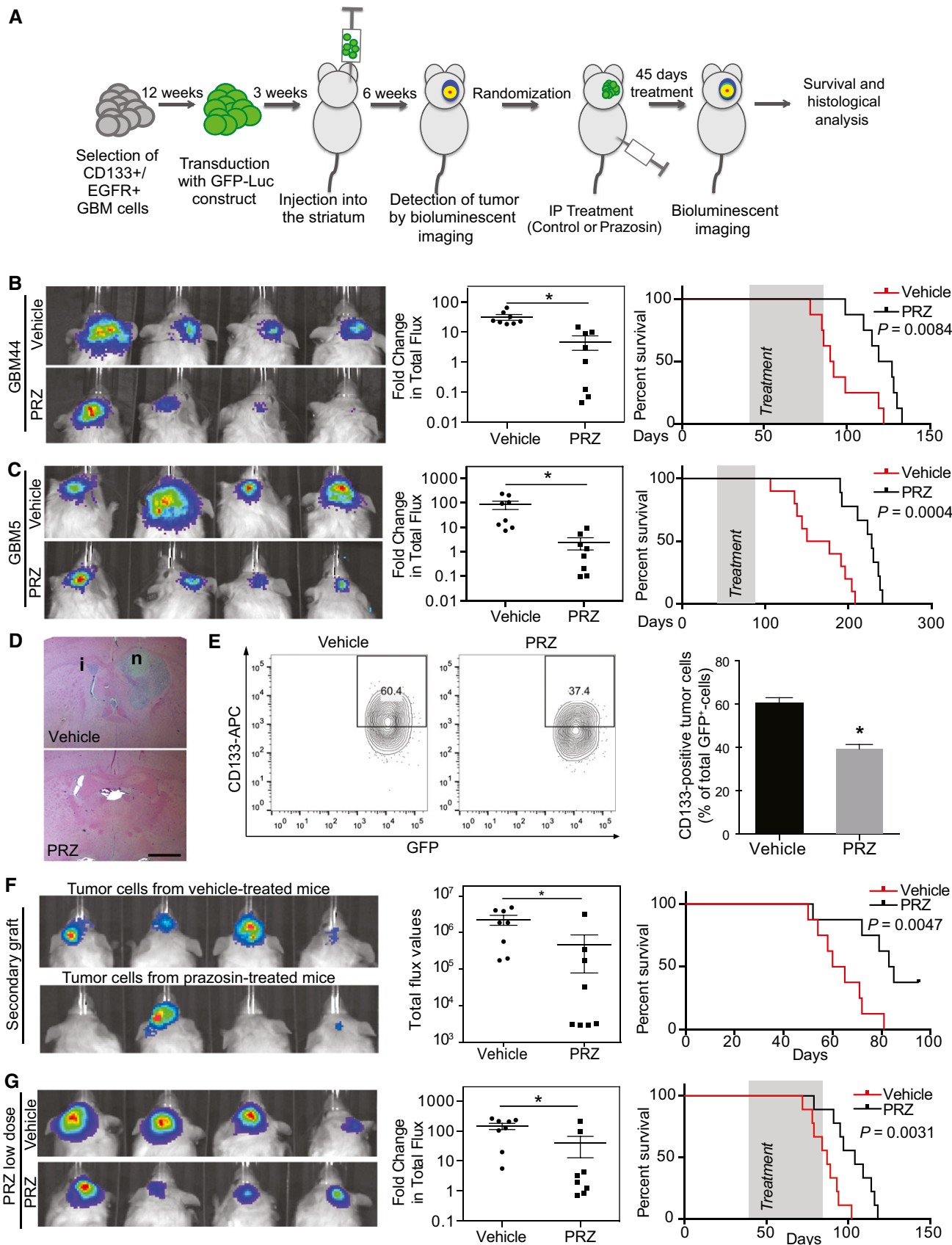

**Figure 2.**

**Figure 2.  Prazosin inhibits glioblastoma growth *in vivo*.**

A     Schematic representation of prazosin treatment *in vivo*. EGFR[+]/CD133[+] GICs directly isolated from primary human glioblastoma samples (GBM44 and GBM5) were transduced with a GFP-luciferase construct and orthotopically implanted into the striatum of NSG mice. Treatment was initiated once the tumors were detected and bioluminescent analyses of tumor growth were performed after 45 days of treatment.

B, C   *In vivo* effect of prazosin treatment (1.5 mg/kg) on glioblastoma growth. Tumors were initiated with GBM44 GICs (B) or GMB5 GICs (C). Left panels: Bioluminescent *in vivo* images of tumors in mice treated with prazosin (PRZ) or vehicle for 45 days. Middle panels: Quantification of the bioluminescent signals. Fold change in total flux represents the ratio: total flux after treatment/total flux before treatment. *$P$ = 0.0002 and *$P$ = 0.003 for GBM44 and GBM5, respectively, $n$ = 8, two-sided Mann–Whitney $U$-test. Right panels: Kaplan–Meyer survival curves of mice treated with prazosin (PRZ) or vehicle demonstrating a significant survival benefit of prazosin as compared to vehicle, log-rank Mantel–Cox test. The treatment period is shaded in gray.

D     Example of hematoxylin/eosin staining of brain coronal sections from mice sacrificed after 45 days of treatment with prazosin (PRZ) or vehicle. i: tumor infiltration. n: tumor necrosis. Scale bar: 2 mm. GBM44 GICs.

E     Left panel: Representative flow cytometry plots depicting the percentage of CD133[+] glioblastoma cells (GFP[+]) isolated from mice treated with prazosin (PRZ) or vehicle. Tumors were initiated with GBM44 GICs. Right panel: Quantification of flow cytometry analyses of CD133 expression by GFP[+]-tumor cells isolated from xenografts of three prazosin-treated (PRZ) mice and three vehicle-treated mice. *$P$ = 0.0003, $n$ = 6, two-sided Mann–Whitney $U$-test. Results are presented as mean ± SD in biological quadruplicates from six independent experiments.

F     Secondary grafting of tumor cells (GBM44-GFP[+]) isolated from vehicle- or prazosin-treated mice bearing primary tumors. Mice injected with glioblastoma cells isolated from prazosin-treated mice developed tumors at a lower frequency (50% versus 100%). Moreover, mice injected with glioblastoma cells isolated from prazosin-treated mice presented a statistically significant survival benefit. Left panel: Bioluminescent *in vivo* images of secondary grafts. Middle panel: Quantification of the bioluminescent signals. *$P$ = 0.0004, $n$ = 8, two-sided Mann–Whitney $U$-test. Right panel: Kaplan–Meyer survival curves of mice bearing secondary graft from glioblastoma cells isolated from vehicle- and prazosin-treated mice.

G     Inhibitory effect of low doses of prazosin (0.15 mg/kg) on glioblastoma growth *in vivo*. Left panel: Bioluminescent *in vivo* images of tumors in mice treated with vehicle or prazosin (PRZ) for 45 days. Tumors were initiated with GBM44 GICs (compare with panel B). Middle panel: Quantification of the bioluminescent signals. *$P$ = 0.0007, $n$ = 8, two-sided Mann–Whitney $U$-test. Right panel: Kaplan–Meyer survival curves of mice demonstrating a significant survival benefit of a treatment with low doses of prazosin, log-rank Mantel–Cox test. The treatment period is shaded in gray.

constitute a population of GICs with a high degree of self-renewal and tumor-initiating ability (Mazzoleni *et al*, 2010; Emlet *et al*, 2014), were sorted from primary patient-derived glioblastoma samples and transduced with a GFP-luciferase construct prior to injection into mice brains for further bioluminescence imaging. Forty-five-day-long treatments were initiated once the presence of tumor masses was confirmed by *in vivo* bioluminescent imaging (Fig 2A). Prazosin inhibited glioblastoma growth compared to control in both xenograft models (Fig 2B–D), and Kaplan–Meier analysis showed a significant improvement in survival of the groups treated with prazosin as compared to the control groups (Fig 2B and C). Histological analysis performed at the end of the treatment period confirmed that prazosin-treated mice presented smaller tumors than vehicle-treated mice (Fig 2D). Of note, tumors from vehicle- and prazosin-treated mice presented similar blood vessels density, suggesting that prazosin did not affect angiogenesis (Fig EV1C). Flow cytometry analysis of GFP-positive tumor cells showed a significant decrease in human CD133-positive cells in prazosin-treated mice, suggesting removal of GICs along with the non-GICs (Fig 2E). To further demonstrate that prazosin affects GICs, we evaluated its effects on a major property of cancer stem cells, tumor initiation. GFP-positive tumor cells from primary tumors were isolated (see Materials and Methods section) and re-injected into new groups of mice (Fig 2F). All mice that were grafted with glioblastoma cells isolated from vehicle-treated mice developed tumors (8/8 cases, Fig 2F). However, only 4/8 mice injected with glioblastoma cells isolated from prazosin-treated mice developed tumors (Fig 2F). Moreover, mice injected with glioblastoma cells isolated from prazosin-treated mice showed a statistically significant survival benefit ($P$ = 0.0047) (Fig 2F). We also tested lower doses of prazosin (0.15 mg/kg instead of 1.5 mg/kg) compatible with the human daily regimen for treatment of hypertension (see Discussion section). The lower dose of prazosin also induced a significant reduction in tumor growth and increased survival of glioblastoma-bearing mice (Fig 2G). To verify whether prazosin effects could also be observed in an immunocompetent syngeneic mouse model, we implanted the mouse glioblastoma-like cell line GL261, transduced with GFP-luciferase, in C57Bl/6 mice brains. Prazosin induced GL261 cell death *in vitro* (Fig 3A) and significantly inhibited tumor growth *in vivo* (Fig 3B–D), an effect associated with a survival benefit (Fig 3C). Finally, using this glioblastoma model coupled with intraperitoneal injections of the green-fluorescent derivative of prazosin, BODIPY FL prazosin, we observed a marked accumulation of prazosin in the tumor within 2 h post-treatment (Fig 3E). Taken altogether, these data show that prazosin inhibits tumor growth initiated by GICs *in vivo* and increases the survival of glioblastoma-bearing mice including at low doses akin to those used in human treatments.

## Prazosin-induced GIC apoptosis is independent from α-AR

We further explored prazosin mechanism of action in GICs. We first observed that prazosin induced caspase-3 activation in TG1 cells, and this effect was blocked by ZVAD, a caspase inhibitor (Fig 4A). Interestingly, prazosin did not activate caspase-9, suggesting that it does not target the intrinsic pathway of apoptosis activated by cytotoxic agents such as etoposide (Fig 4B). Prazosin-induced GIC apoptosis was also observed by FACS analysis of Annexin V and DAPI staining (Fig EV1A). ZVAD prevented prazosin-induced GIC death, confirming that prazosin triggers GIC apoptosis (Fig 4C). We further confirmed that prazosin triggers glioblastoma cells apoptosis *in vivo*, as glioblastoma cells (GFP-positive) isolated from prazosin-treated mice presented higher levels of Annexin V and DAPI staining compared to glioblastoma cells isolated from vehicle-treated mice (Fig 4D). Of note, prazosin did not induce apoptosis in non-tumor stromal (GFP-negative) cells (Fig 4D), further confirming the absence of toxicity of this drug on normal cells. Using TUNEL staining, we also observed increased numbers of cells undergoing apoptosis following *in vivo* prazosin treatment of mice bearing tumors initiated by GBM44 grafting (Fig EV1B).

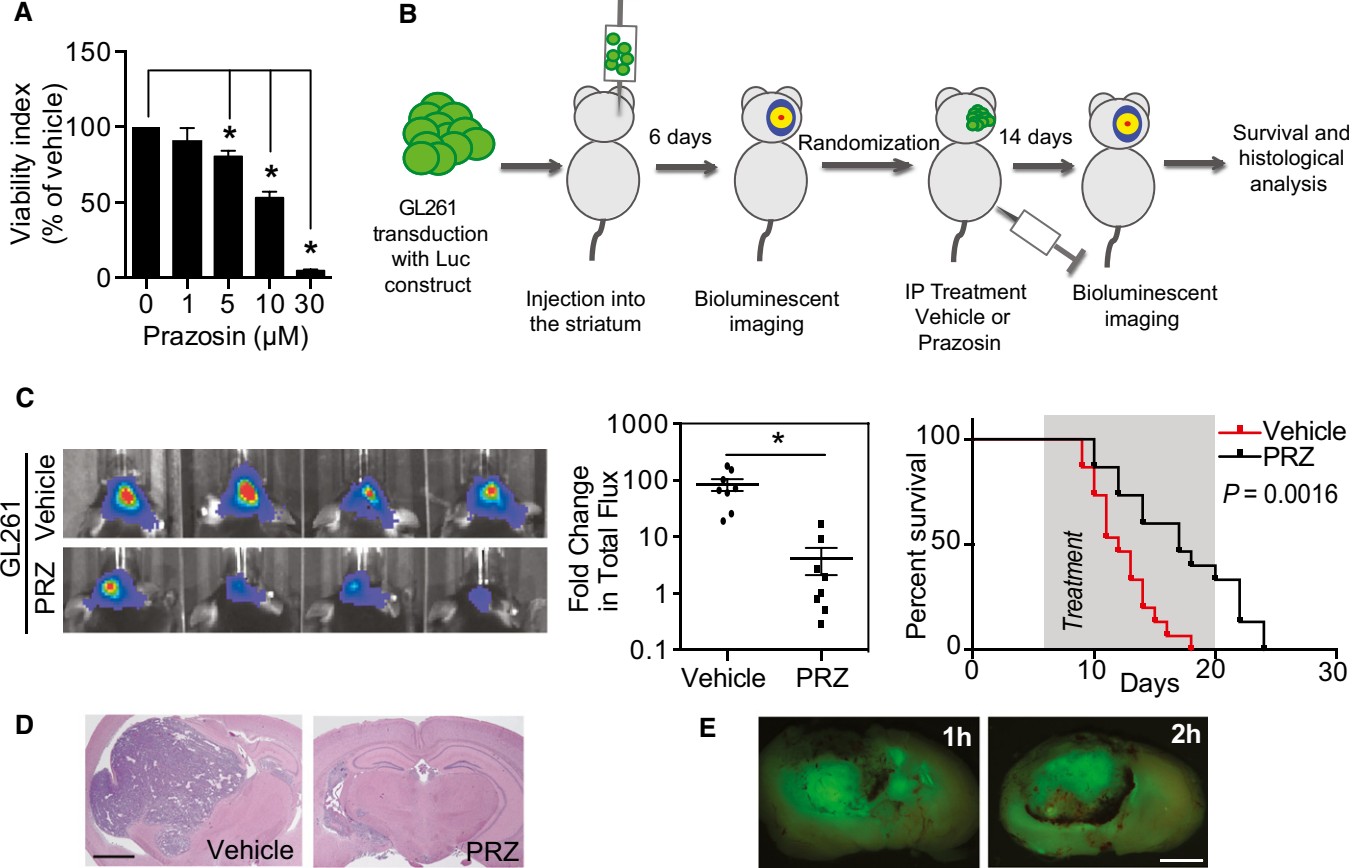

**Figure 3. Prazosin inhibits glioblastoma growth in immunocompetent mice.**

A  Viability analysis of GL261 treated with prazosin *in vitro*. *P = 0.0286, n = 4, two-sided Mann–Whitney *U*-test. Results are presented as mean ± SD in biological quadruplicates from six independent experiments.

B  Schematic representation of prazosin treatment *in vivo* of glioblastoma-bearing immunocompetent mice.

C  Prazosin inhibits *in vivo* tumor growth in an immunocompetent model of glioblastoma. Left panel: Bioluminescent *in vivo* images of tumors in C57Bl/6 mice treated with vehicle or prazosin (PRZ) for 14 days. Middle panel: Quantification of the bioluminescent signals. *P = 0.0002, n = 8, two-sided Mann–Whitney *U*-test. Right panel: Kaplan–Meyer survival curves of mice treated with prazosin (PRZ) or vehicle. Log-rank Mantel–Cox test. The treatment period is shaded in gray.

D  Hematoxylin/eosin staining of C57Bl/6 mice brain coronal sections sacrificed after 14 days of treatment with prazosin. Scale bar: 2 mm.

E  Fluorescence imaging of GL261 intra-striatal grafts at 1 and 2 h after an intraperitoneal injection of BODIPY FL prazosin, the green-fluorescent derivative of prazosin. Scale bar: 2 mm.

Cell cycle was mostly not affected by prazosin. Although we observed a dose-dependent reduction in BrdU incorporation *in vitro* in GICs that had survived a 24-h prazosin exposure, and a decrease in Ki67 staining in tumor grafts of prazosin-treated mice (Fig EV3A and B), no change was observed in cyclin D1, cyclin D3, and CDK2 levels, which are required for G1/S transition (Fig EV3C).

Prazosin is known as a non-selective antagonist of α1-AR and as a selective antagonist of α2B-AR (Bylund *et al*, 1994). Prazosin inhibited GIC survival in a concentration-dependent manner, with an $EC_{50}$ value of 7.88 μM (6.70–9.28) (Fig 4E), several orders of magnitude above the nanomolar concentrations at which prazosin acts on α-ARs (http://www.bindingdb.org/bind/ByLigandName.jsp). Having observed that the α2B-AR antagonists terazosin and ARC 239 did not mimic prazosin effect on GIC viability (Fig 1A), we sought to test whether cirazoline, a non-selective subtype agonist of α1-AR, would reverse prazosin-induced GIC death. Cirazoline did not affect GIC viability, and only partially rescued prazosin-induced

cell death (Fig 4F). Accordingly, radio ligand assays showed that prazosin binding sites were not detected in GIC membrane preparations (Fig 4G). Considering the reported stimulation of the ERK/MAPK pathway by α-AR ligands (Benoit *et al*, 2004; Liou *et al*, 2009), and the reported participation of this signaling pathway in the promotion of GIC survival (Sunayama *et al*, 2010), we analyzed the effect of prazosin on ERK/MAPK pathway in GICs. In contrast to the expected effect for an α-AR antagonist, prazosin induced ERK phosphorylation in these cells (Fig 4H). ERK/MAPK pathway activation in GICs was confirmed by SRE-luciferase reporter assay, which is a measure of luciferase reporter expression driven by the serum response element downstream of the ERK/MAPK pathway (Fig 4I). Finally, the ERK/MAPK pathway inhibitor U0126 did not protect the GICs from prazosin (Fig 4J), confirming that prazosin-induced GIC death was independent of ERK/MAPK pathway activity. Altogether, these results indicate that prazosin's effect on GICs is independent of AR- and ERK-/MAPK-signal transduction pathway.

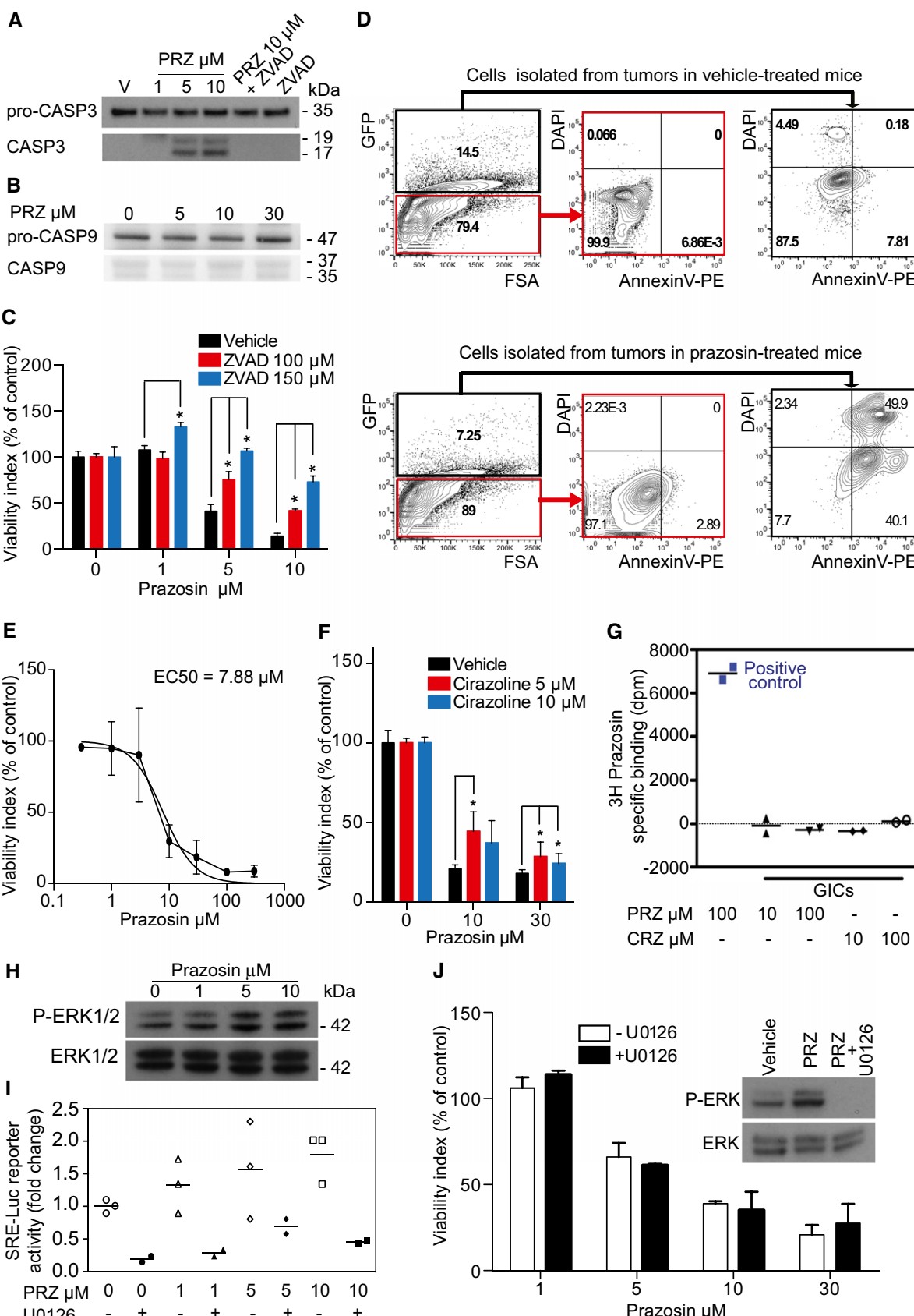

**Figure 4.**

**Figure 4. Prazosin induces GIC apoptosis independently from α-AR.**

A   Immunoblotting for pro-caspase-3 (pro-CASP3) and active caspase-3 (CASP3) in GICs treated with prazosin (PRZ) or vehicle (V) demonstrating that prazosin activates caspase-3. ZVAD, a caspase inhibitor, prevents prazosin-induced caspase-3 activation. kDa: kilodaltons.

B   Immunoblotting for pro-caspase-9 (pro-CASP9) and active caspase-9 (CASP9) in GICs treated with prazosin (PRZ) or vehicle (V) demonstrating that prazosin does not activate caspase-9.

C   Viability analysis of GICs treated with prazosin for 24 h in the presence or absence of ZVAD, a caspase inhibitor. ZVAD counteracts prazosin-induced GIC death. *P = 0.0286, n = 4, two-sided Mann–Whitney U-test. Results are mean ± SD in biological quadruplicates from three independent experiments.

D   Prazosin induces glioblastoma cell apoptosis *in vivo*. GFP$^+$ and GFP$^-$ cells were analyzed from tumors in vehicle (upper panel) or prazosin-treated (lower panel) mice. Annexin V/DAPI staining was used to identify apoptotic cells by FACS. Tumors were initiated by GBM44 GICs implantation.

E   Dose–response curve of prazosin on GIC survival (24 h treatment). EC$_{50}$ was determined with curve fit using nonlinear regression. Results are mean ± SD in biological triplicates from one experiment out of three independent experiments giving similar results.

F   Viability analysis of GICs treated with prazosin for 24 h in the presence or absence of cirazoline, a subtype agonist of the α-ARs. Cirazoline did not alter GIC survival and counteracted only poorly prazosin-induced GIC death. *P < 0.05, n = 4, two-sided Mann–Whitney U-test. Results are mean ± SD in biological quadruplicates from three independent experiments.

G   Membrane binding assays demonstrating the absence of prazosin binding sites in GIC membrane preparations. Positive control shows that prazosin binds to membrane preparations of yeast expressing α1-AR. PRZ: prazosin. CRZ: cirazoline.

H   Immunoblotting for phosphorylated ERK1/2 (P-ERK1/2) and total ERK1/2 following prazosin treatment for 30 min. Prazosin induces ERK1/2 phosphorylation in GICs.

I   SRE-luciferase reporter activity analysis of GICs treated with prazosin in absence or presence of U0126 (10 μM), an inhibitor of the ERK-activating kinase MEK. U0126 prevented prazosin-induced SRE-luciferase activation in GICs. Results are from three independent experiments.

J   Viability analysis of GICs treated with prazosin for 24 h in the presence or absence of U0126 (10 μM), an inhibitor of the ERK-activating kinase MEK. U0126 did not counteract prazosin-induced GIC death. Blockade of prazosin-induced ERK1/2 phosphorylation by U0126 was confirmed by immunoblotting (insert). PRZ: prazosin. Results are presented as mean ± SD in biological quadruplicates from three independent experiments.

Source data are available online for this figure.

## PKCδ is involved in prazosin-induced GIC apoptosis

PKCδ is a member of the family of novel protein kinase C (nPKC) isoforms, which has been shown to promote apoptosis by acting both upstream and downstream of caspase-3 (Basu *et al*, 2001; Lu *et al*, 2007). Immunoblotting showed that GICs expressed higher levels of PKCδ compared to NSCs (Fig 5A). Of note, GICs and glioblastoma cells that differentiated along the astroglial, oligodendroglial, and neuronal lineages expressed similar levels of PKCδ (Fig EV4A). Immunocytochemical detection of PKCδ in TG1 cells revealed enhanced punctate nuclear staining in prazosin-treated GICs, as compared to vehicle-treated GICs (Fig 5B). As caspase-3 activation is known to induce PKCδ cleavage into a 41 kDa catalytic fragment (Basu *et al*, 2001; Lu *et al*, 2007), we investigated the

presence of PKCδ catalytic fragment in prazosin-treated GICs. Immunoblotting analysis revealed that prazosin induced the cleavage of PKCδ into its 41 kDa catalytic fragment (Fig 5C). Rottlerin, a selective PKCδ inhibitor (Gschwendt *et al*, 1994), prevented prazosin-induced PKCδ cleavage in GICs (Fig 5C). Moreover, prazosin-induced caspase-3 activation in GICs was also prevented by rottlerin (Fig 5D) suggesting that PKCδ activation is necessary for the processing of caspase-3 in GICs. We then investigated whether the inhibition of PKCδ could rescue GICs from prazosin-induced cell death. In addition to rottlerin, we used the peptide δV1.1 that specifically opposes PKCδ mobilization (Chen *et al*, 2001) and silenced the expression of PKCδ using shRNA (Fig EV4B). Rottlerin (Fig 5E), δV1.1 (Fig 5F), and PKCδ shRNA (Fig 5G) significantly rescued GICs from prazosin-induced glioblastoma cell death.

**Figure 5. Prazosin induces GIC apoptosis in a PKCδ-dependent manner.**

A   PKCδ expression analysis in two patient-derived GICs and NSCs by immunoblotting. GICs express higher levels of PKCδ than NSCs. kDa: kilodaltons.

B   PKCδ detection by immunostaining (green). Nuclei were stained with DAPI (blue). Enhanced PKCδ punctate nuclear expression in prazosin-treated (PRZ, 5 μM) GICs, as compared to vehicle-treated GICs. PRZ: prazosin. Scale bar: 10 μm.

C   Total PKCδ (PKCδ 78 kDa) and cleaved PKCδ (PKCδ 41 kDa) expression analysis in GICs by immunoblotting. Prazosin (PRZ) induces PKCδ cleavage into a 41 kDa active fragment. 2 μM rottlerin (Rott), a PKCδ inhibitor, inhibits prazosin-induced PKCδ cleavage. V: vehicle. kDa: kilodaltons.

D   Pro-caspase-3 (Pro-CASP3) and activated caspase-3 (CASP3) expression analysis in GICs by immunoblotting. 2 μM rottlerin (Rott) inhibits prazosin (PRZ)-induced caspase-3 activation. V: vehicle. kDa: kilodaltons.

E   Inhibition of PKCδ by rottlerin prevents, in a concentration-dependent manner, prazosin-induced GIC death. Viability analysis of GICs treated with prazosin for 72 h in the presence or absence of rottlerin. *P = 0.0286, n = 4, two-sided Mann–Whitney U-test. Error bars mean ± SD.

F   Inhibition of PKCδ by the specific anti-PKCδ RACK peptide δV1.1 (10 μM) counteracts prazosin-induced GIC death. Viability analysis of GICs treated with prazosin for 72 h in the presence or absence of δV1.1. *P < 0.005 for prazosin 1 and 5 μM and P < 0.001 for Prazosin 10 μM by two-tailed unpaired Student's t-test, n = 4. Error bars mean ± SD.

G   Decreased PKCδ expression using shRNA counteracts prazosin-induced GIC death. Viability analysis of GICs transduced with scrambled or PKCδ shRNA and treated with prazosin for 72 h. *P < 0.005 by two-tailed unpaired Student's t-test, n = 4. Error bars mean ± SD.

H   Prazosin inhibits AKT phosphorylation in GICs. LY294002 (LY, 30 μM), an inhibitor of the PI3K/AKT pathway, was used as a positive control. Terazosin, which does not affect GIC viability, does not alter AKT phosphorylation (see the corresponding cell viability counting in Fig EV4C). Phosphorylated AKT (P-AKT) and total AKT (AKT) expression analysis in GICs by immunoblotting. V: vehicle. kDa: kilodaltons.

I   Prazosin (10 μM) does not inhibit AKT phosphorylation in NSCs. Analysis by immunoblotting. V: vehicle. kDa: kilodaltons.

J   β-catenin expression, a downstream target of AKT, is decreased in prazosin-treated GICs, as opposed to NSCs. Analysis by immunoblotting. V: vehicle. kDa: kilodaltons.

K, L   Inhibition of PKCδ using rottlerin (Rott, 2 μM) or δV1.1 (10 μM) counteracts prazosin inhibition of AKT phosphorylation. Analysis by immunoblotting. V: vehicle. kDa: kilodaltons.

Source data are available online for this figure.

   

As novel PKC isoforms have been shown to curtail AKT activation (Li *et al*, 2006), a signaling pathway reported to be crucial to GIC survival (Eyler *et al*, 2008; Bleau *et al*, 2009), we examined the effect of prazosin on PI3K/AKT pathway in GICs. Prazosin inhibited AKT phosphorylation in GICs as efficiently as LY294002, a specific PI3K inhibitor (Fig 5H, left panel). Terazosin, used as a control

since it does not affect GICs survival (Fig EV4C), did not modify AKT phosphorylation (Fig 5H, right panel). In addition, prazosin did not alter the levels of phosphorylated AKT (P-AKT) in NSCs (Fig 5I). To further investigate prazosin effects on AKT activity, we analyzed the levels of β-catenin expression, one of AKT downstream targets. Activated AKT is known to phosphorylate and thereby

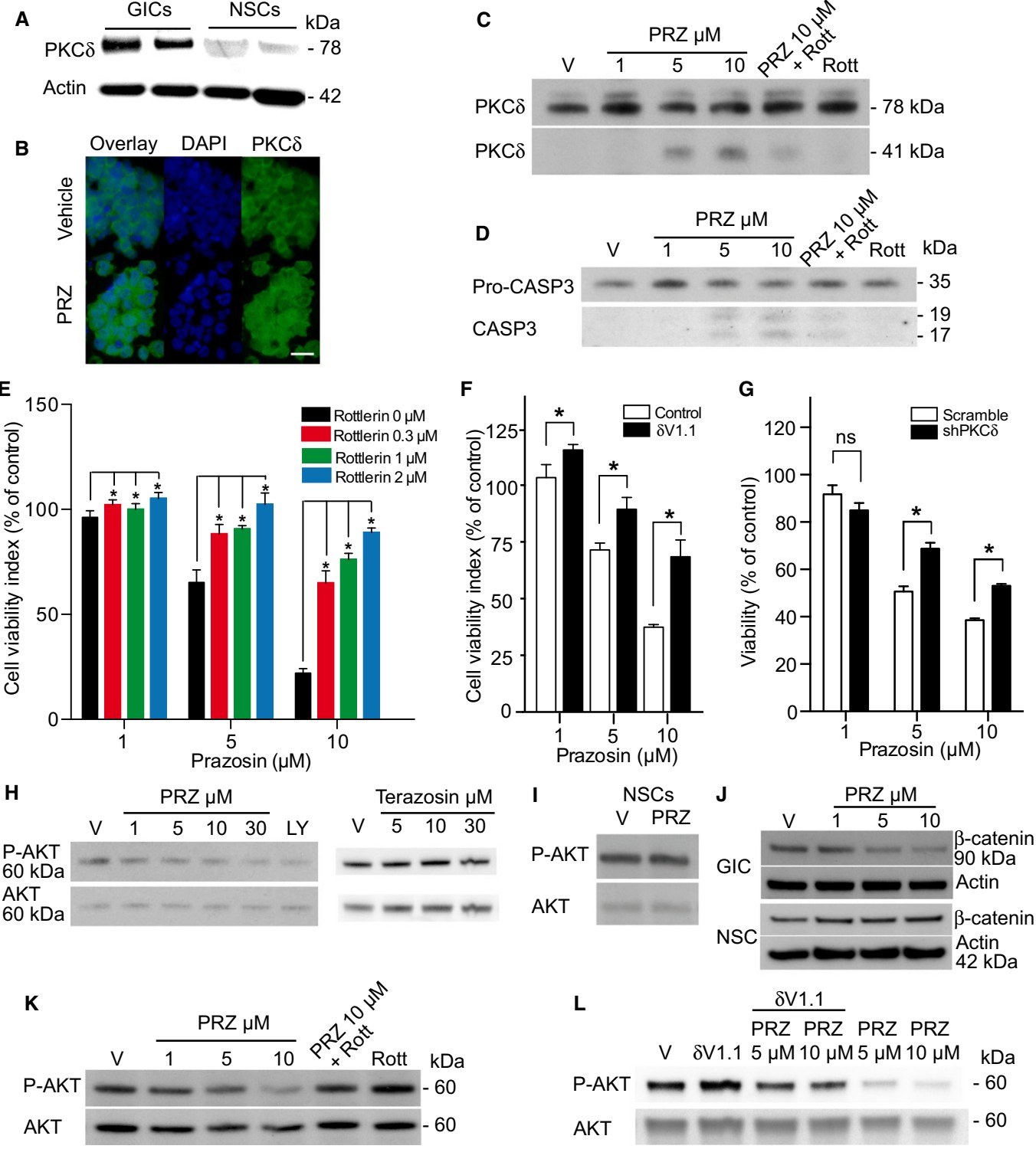

**Figure 5.**

**Figure 6. Prazosin inhibits glioblastoma growth *in vivo* in a PKCδ-dependent manner.**

A   Schematic representation of the experimental design.
B   Bioluminescent *in vivo* images of tumors in mice grafted with GICs expressing either scrambled (upper panel) or PKCδ (lower panel) shRNA. All mice were treated with prazosin (PRZ) for 45 days.
C   Quantification of the bioluminescent signals showing that decreased expression of PKC counteracts prazosin inhibition of tumor growth. Fold change in total flux represents the ratio: total flux after treatment/total flux prior treatment. *$P = 0.0007$, $n = 8$, two-sided Mann–Whitney U-test.
D   Schematic representation of the molecular mechanisms underlying prazosin-induced GIC apoptosis. See text for details.

inactivate GSK-3β, leading to stabilization of the transcription factor β-catenin (Yost *et al*, 1996; Pap & Cooper, 1998). As expected, prazosin-treated GICs showed reduced β-catenin levels as compared to vehicle-treated GICs (Fig 5J). Conversely, prazosin did not affect β-catenin levels in NSCs (Fig 5J). We then exposed GICs to prazosin in the presence or absence of rottlerin (Fig 5K) or δV1.1 (Fig 5L) and assessed the levels of P-AKT. Rottlerin as well as δV1.1 counteracted prazosin-induced inhibition of AKT phosphorylation (Fig 5K and L). The role of PKCδ was further investigated *in vivo*. Silencing the expression of PKCδ with shRNA (Fig EV4B) slowed the growth of the tumors (Fig 6B, Prior-PRZ images). Interestingly,

analysis of mRNA profiles of adult glioblastoma available in the TCGA dataset showed that high expression of PKCδ is associated with a poorer prognosis for patients. High PKCδ (*PRKCD*) mRNA levels were inversely correlated with overall survival as well as progression-free survival (Fig EV5). This result suggests that PKCδ expression is necessary for GIC tumor growth *in vivo*, and is consistent with the reported inhibitory effects of PKCδ shRNA on the growth of xenografts of prostate, pancreas, and breast cancer stem cells (Chen *et al*, 2014). We then analyzed whether prazosin could still inhibit the growth of PKCδ-silenced tumors *in vivo*. As expected, prazosin inhibited the growth of control (shScramble) tumors

(Fig 6B, upper panel). On the other hand, PKCδ-silenced tumors were no longer responsive to prazosin treatment (Fig 6B, lower panel and Fig 6C), further confirming the involvement of PKCδ in prazosin-induced glioblastoma cell death.

Taken together, these results show that prazosin-induced PKCδ activation leads to AKT pathway inhibition. PKCδ-dependent AKT inhibition is accompanied by caspase-3 activation, generating a PKCδ catalytic fragment, ultimately leading to GIC apoptosis (Fig 6D).

## Discussion

Comparative analysis of α-AR antagonists on cell survival using patient-derived GICs showed that only prazosin inhibited GIC viability in a robust and concentration-dependent manner. Moreover, prazosin inhibited the growth of glioblastoma in orthotopic xenograft mouse models and increased mice survival, with no toxicity. We demonstrate that prazosin-induced GIC apoptosis involves a PKCδ-dependent inhibition of AKT pathway.

The α-AR family includes three α1 and three α2 subtypes. Prazosin has been extensively characterized as a non-selective subtype antagonist of α1-AR and a selective subtype antagonist of α2B-AR (Bylund *et al*, 1994). We found that prazosin acted on GICs independently from ARs. Membrane preparations of GICs did not show prazosin binding sites. Cirazoline, a non-selective subtype agonist of α-AR did not alter GIC survival, and mildly counteracted prazosin-induced GIC death. In addition, prazosin activated the MAPK/ERK signaling pathway in GICs, an observation that contrasts with the expected inhibitory effect of α-AR blockage on this signaling pathway (Benoit *et al*, 2004; Liou *et al*, 2009). Altogether, these data demonstrate an off-target mechanism of action for prazosin on GICs.

Caspase-3 activation in GICs exposed to prazosin, and prevention of prazosin-induced GIC death by ZVAD, an inhibitor of caspase activation, revealed that the type of death induced by prazosin in GIC is apoptosis. Other quinazoline-related compounds such as doxazosin and terazosin, shown here to only marginally affect GICs, have been reported to exert pro-apoptotic effects on prostate cancer cells independently from ARs, through the activation of TGFβ signaling pathway (Desiniotis & Kyprianou, 2011). A pro-apoptotic action of prazosin has been reported in the K562 erythroleukemia cell line, where prazosin has been shown to enter the cells and bind intracellular proteins (Fuchs *et al*, 2011). The reason why prazosin is the most effective of the quinazolines on GICs remains to be determined. Interestingly, a preferential prazosin toxicity, among other quinazoline-based alpha-adrenoreceptor antagonists, has been reported on prostate cancer cells, 10 μM prazosin being in this case more effective than 100 μM doxazosin in inducing DNA damage (Lin *et al*, 2007).

We describe here a novel mechanism where prazosin-induced GIC apoptosis includes a mechanism dependent on PKCδ activation, a critical pro-apoptotic signal in various cell types (Leverrier *et al*, 2002; Brodie & Blumberg, 2003; Reyland, 2007; Larroque-Cardoso *et al*, 2013). Whether prazosin binds directly to PKCδ remains to be determined. PKCδ is known to promote apoptosis through its constitutively activated form, the 41 kDa PKCδ catalytic fragment. Generation of the PKCδ catalytic fragment depends on caspase-3 activation.

Caspases are the main effectors of apoptosis and are activated in response to apoptotic stimuli by limited proteolytic cleavage, an event prevented by AKT activation (Salvesen & Riedl, 2008). AKT is a serine–threonine kinase that regulates cell survival in various types of cells (Parcellier *et al*, 2008), including GICs (Eyler *et al*, 2008; Bleau *et al*, 2009). Our results show that the effect of prazosin on GICs depends on the inhibition of the AKT signaling pathway, which may occur in response to PKCδ activation (Fig 5). PKCδ-dependent AKT inhibition is accompanied by the activation of caspase-3, a downstream target of AKT. Blockade of PKCδ prevented prazosin-induced caspase-3 activation, suggesting that PKCδ activation is required for the processing of caspase-3. Moreover, prazosin-induced GIC apoptosis is mostly dependent on PKCδ activation, resulting in AKT signaling pathway inhibition. Altogether, our results demonstrate PKCδ as a novel protagonist in GIC apoptosis. In addition, these data may improve the prospects for targeting other cancer cells known to exhibit altered PKCδ signaling (Bosco *et al*, 2011).

Prazosin effects were observed not only *in vitro*, but also *in vivo*, using orthotropic implants of patient-derived GICs. The importance of targeting GICs stems from repeated descriptions of the increased tumor-seeding potential of these cells and their resistance to a variety of chemotherapy and radiation treatments, as compared to the other tumor cell populations (Eyler & Rich, 2008; Zhu *et al*, 2014). Failure of current therapies to eradicate the disease is likely caused by the replenishment of the tumor by GICs spared by the treatment (Chen *et al*, 2012a). Development of effective treatments targeting GICs requires not only understanding the molecular mechanism of action of the novel compound but also determining the drug's safety in patients. Our data demonstrate that prazosin is highly active on GICs and their differentiated progeny, both *in vitro* and *in vivo*, whereas it marginally affects the survival of NSCs, or NSCs differentiated along either of the three neural lineages. Although the signaling pathways sustaining maintenance of NSCs and GICs differ in several ways, they both require a proper functioning of the PI3K/ AKT pathway for their survival (Groszer *et al*, 2006; Yan *et al*, 2013). Accordingly, no change in AKT phosphorylation was observed in NSCs following prazosin treatment. This result, associated with the paucity of PKCδ levels in NSCs as compared to GICs, suggests that a preliminary activation of PKCδ is mandatory for prazosin to exert its pro-apoptotic action. The possibility that additional molecular mechanisms are involved in prazosin-induced cell death cannot be excluded but remains to be elucidated. An effect of prazosin on adult human neural cells cannot be excluded since we used human NSCs of embryonic origin, although no deleterious effect of prazosin was observed (see Figs 2D, 3D, and 4D) or has been reported so far in mouse brain following administration of doses akin to the ones we used.

Prazosin is a clinically approved drug and has been adopted for more than 40 years in clinical practice to treat hypertension (Cavero & Roach, 1980). Its use has been extended to treat benign prostatic hyperplasia, congestive heart failure, pheochromocytoma, sleep problems associated with post-traumatic stress disorder, and Raynaud's disease (http://www.ncbi.nlm.nih.gov/pubmedhealth/ ?term = prazosin). The inhibition of glioblastoma growth *in vivo* and extended survival of glioblastoma-bearing mice in the presence of prazosin support the use of prazosin as an adjuvant treatment for glioblastoma patients.

The current use of prazosin hydrochloride in humans as an oral prescription for hypertension is, according to the FDA recommendation, a total daily dose of 20 mg that may be further increased up to 40 mg given in divided doses. Bioavailability studies have demonstrated peak levels of approximately 65% of the drug in solution (http://www.fda.gov/Safety/MedWatch/SafetyInformation/Safety-RelatedDrugLabelingChanges/ucm155128.htm). The daily dosage already approved is therefore likely to result in a bioavailability of prazosin in the 5–10 μM range, shown here to effectively induce GIC apoptosis *in vitro*. Such a potential use as adjuvant therapy in humans is further supported by our observations that low doses of prazosin (0.15 mg/kg should be extrapolated to 9 mg for an adult of 60 kg) also significantly inhibited tumor growth *in vivo*.

Prazosin-induced patient-derived GIC apoptosis and inhibition of glioblastoma growth *in vivo* through a PKCδ-dependent mechanism of action, associated with the well-established and documented use of this drug in clinical settings, opens a promising possibility for its use in glioblastoma treatment.

## Materials and Methods

### Cell culture

Patient-derived GICs (TG1, TG16, GBM5, GBM44) were isolated from glioblastoma neurosurgical resections, and their stem-like properties (i.e., clonogenicity, self-renewal, expression of neural stem cell markers) and tumor-initiating properties were previously characterized (Patru *et al*, 2010; Galan-Moya *et al*, 2011; Silvestre *et al*, 2011; Fareh *et al*, 2012; Emlet *et al*, 2014). All cell cultures were authenticated by STR profiling, and absence of mycoplasma was controlled. Sequencing RT–PCR products of *TP53* and *PTEN* transcripts showed the wild-type forms of both genes in TG1 GICs and mutant forms in TG16 GICs (Silvestre *et al*, 2011). Neural stem cells/neural progenitor cells (NSC24, NSC25, NSC5031, NSC8853) were derived from electively terminated human fetal brains, and characterized as previously described (Thirant *et al*, 2011). NSCs expressed the neural stem cell markers Sox2, Bmi1, and Nestin (Thirant *et al*, 2011). GICs and NSCs were cultured as floating cellular spheres as previously described (Patru *et al*, 2010; Thirant *et al*, 2011; Feve *et al*, 2014).

### Chemicals and antibodies

ARC 239 dihydrochloride (selective α2B-AR antagonist), BMY 7378 dihydrochloride (high-affinity α1D-AR antagonist), doxazosin mesylate (selective α1-AR antagonist), prazosin hydrochloride (nonselective α1 and selective subtype α2B-AR antagonist), terazosin hydrochloride (selective α1-AR and α2B-AR antagonist), and cirazoline (selective α1-AR agonist) were purchased from Tocris Cookson Inc. U0126 (MEK inhibitor), LY294002 (PI3K inhibitor), and ZVAD (caspase inhibitor) were purchased from Calbiochem. Rottlerin (PKCδ inhibitor) was from Sigma (Lyon, France). Delta V1.1 was kindly provided from Daria Mochly group (Chen *et al*, 2001). BODIPY® FL Prazosin was from Thermo Scientific. The antagonists and the inhibitors were dissolved in dimethyl sulfoxide (DMSO, Sigma) or water, according to the manufacturer's instructions. Cells were treated with a single dose of antagonists, inhibitors, or the

corresponding volume of vehicle. Inhibitors were added 30 min before the addition of prazosin.

Primary antibodies against the following proteins were used: PKCδ (Santa Cruz Biotechnologies), phospho-p42/p44 ERK (Cell Signaling), p42/p44 ERK, β-tubulin (Millipore), phospho-AKT, AKT, cleaved caspase-3, caspase-3, cyclin D1, cyclin D3, CDK2 (Cell Signaling), β-catenin (BD Biosciences), α-actin (Chemicon International), CD133/1-APC, CD133/2-APC (Miltenyi Biotech), GFAP (Synaptic Systems), O4 (R&D), Ki67 (Thermo Scientific), and EGFR-Alexa 488 (Biolegend). The secondary antibodies were Alexa 488-conjugated goat anti-rabbit (Life Technologies), anti-mouse IgG, and anti-rabbit IgG (both from GE Healthcare). See Appendix Table S1 for references and dilutions.

### Cell viability and proliferation assays

Cells were plated in 96-well plates at $2 \times 10^4$ cells/well and treated with the appropriate compounds at 37°C, 5% $CO_2$. Cell viability was assessed using reduction in WST-1 (4-[3-(4-iodophenyl)-2-(4-nitrophenyl)-2H-5-tetrazolio]-1,3-benzene disulfonate, Roche, France) to water-soluble formazan. At the end of the incubation period, 10% (v/v) WST-1 was added to the culture media, and the cells were further cultured for 3 h. The absorbance was measured at 430 nm in a microplate reader (Expert Plus V1.4 ASYS). Cell viability was also verified with cell counting following addition of trypan blue (Sigma) at a final concentration of 0.1% (v/v), and routine examination of the cells under phase contrast microscopy. Cell proliferation was assessed using the Cell Proliferation ELISA (Roche), which allows immunochemiluminescent detection of BrdU incorporation into newly synthesized DNA of replicating cells, following the manufacturer's instructions. Light emission was measured using a luminometer (EnVision Multilabel Reader 2104 PerkinElmer). Data were normalized relative to the vehicle-treated controls.

### Limiting dilution assays

Cells were plated in 96-well plates at 1, 5, 10, 20, 50, and 100 cells/well/100 μl as previously described (Patru *et al*, 2010). The percentage of wells with neurospheres was determined after 10 and 21 days. Analysis was performed with software available at http://bioinf.wehi.edu.au/software/elda/ (Hu & Smyth, 2009).

### Membrane binding assays

Cellular spheres were dissociated in phosphate-buffered saline (PBS) and subjected to two rounds of resuspension in 10 ml of cold PBS followed by centrifugation at 210 *g* at 4°C. Cells were then resuspended in 10 ml of lysis buffer composed of 10 mM HEPES, 2 mM EGTA at pH 7.4, in the presence of 1× protease inhibitor cocktail (Roche). After centrifugation at 210 *g*, the pellet was resuspended in 1 ml of lysis buffer and incubated for 20 min on ice. The solution was then homogenized and passed 25 times through a 18G needle. After centrifugation at 3,000 *g* for 10 min, the supernatant was collected and centrifuged at 21,000 *g* for 2 h. The pellet was resuspended in lysis buffer and its protein concentration was determined. Radioligand binding for [³H]prazosin (PerkinElmer, France) was performed in a total volume of 100 μl during a 1-h incubation

at 25°C, using 10 or 100 μg membranes. Experiments were performed in binding buffer containing 30 mM HEPES, 150 mM NaCl, 1 mg/ml BSA, pH 7.7 in the presence of 4 nM [$^3$H]prazosin with or without prazosin or cirazoline. Detection limit is 40–60 receptors/cell. Incubations were terminated by rapid vacuum filtration over Whatman GF/C filters. Membrane preparations of yeast expressing the α1-AR (Andre *et al*, 2006) were used as positive controls.

## Neural lineages differentiation

For oligodendrocytic differentiation, cells were plated onto poly-ornithine-/laminin-coated plates or slides in expansion medium for 24–48 h. Medium was changed to DMEM/F12 supplemented with N2, forskolin (10 nM), FGF2 (10 ng/ml), and PDGF (10 ng/ml) for 5 days. From Day 5, medium was switched to DMEM/F12 with N2, thyroid hormone T3 (30 ng/ml), ascorbic acid (200 μM) all supplements from Stem Cell and PDGF (10 ng/ml) (Miltenyi biotech). On Day 15, PDGF was withdrawn from culture to allow maturation and O4 positive could be detected after 5 weeks of differentiation. For neuronal differentiation, cells were plated on poly-ornithine- and laminin-coated plates or slides in expansion medium without EGF. After 10 days, medium was switched to neurobasal, with N2, B27, and FGF2; 4 days later, FGF2 was withdrawn and 4 days after that, medium was switched to neurobasal supplemented with B27 + A, CNTF, and BDNF (Miltenyi biotech). For astrocytic differentiation, cells were treated with 5% serum. All medium supplements were from Stem Cell.

## Immunocytochemistry

Tumor cell spheres were transferred to polyornithine-coated coverslips and allowed to adhere to the coverslips for 20 min at 37°C, 5% CO$_2$. Cells were fixed with 4% paraformaldehyde for 2 min and rinsed with phosphate-buffered saline (PBS). Immunocytochemical staining was performed as previously described (Patru *et al*, 2010). Immunofluorescence was observed with a fluorescent microscope (Eclipse E800, Nikon, USA). Images were acquired with a digital still camera (DXM 1200, Nikon, http://www.nikoninstruments.com) using Lucia software (Laboratory Imaging, Ltd, http://www.laboratory-imaging.com). Images were prepared using Adobe Photoshop (Adobe Systems).

## TUNEL and Ki67 staining

Brains were harvested, fixed in 4% paraformaldehyde at 4°C for 24 h, followed by 70% ethanol for 24 h at room temperature, and embedded in paraffin. For TUNEL staining, brain sections were stained for cell death using the ApoTag plus Peroxidase *In Situ* Apoptosis Detection Kit (Millipore, S7101) as per the manufacturer's instruction. For Ki67 staining, brain sections were treated with 3% H$_2$O$_2$ for 15 min, followed by Antigen Retrieval for 10 min at 100°C in working Tris–EDTA buffer (Abcam ab93684). Blocking was performed with 10% normal horse serum for 30 min. Then, sections were exposed to Ki67 antibody for 2 h at room temperature followed by goat anti-rabbit-biotinylated, 1:500 for 30 min, StreptAvidin–HRP, 1:500 for 30 min, DAB chromogen, and hematoxylin counterstain. Quantitation was performed using

ImageJ Software on monochrome images by measuring the area of positive staining.

## Immunoblotting

Treated cells were harvested and washed with PBS, and protein extracts were prepared as previously described (Thirant *et al*, 2012). Briefly, cells were lysed in 25 mM PIPES pH 6.8, 1% Triton, 0.5 mM EDTA, 0.5 mM EGTA, 1 mM sodium orthovanadate, 5 mg/ml leupeptin, 5 mg/ml pepstatin, 5 mg/ml aprotinin, 1 mM PMSF (Sigma). Proteins were resolved by 4–12% SDS–PAGE and visualized by immunoblotting with the appropriate antibodies. Signal detection was performed with the ECL+ chemiluminescence detection system (PerkinElmer, France). Densitometric analysis was conducted using ImageJ software.

## Luciferase reporter assays

Cell transfection was achieved with the Amaxa Nucleofector Electroporator (Amaxa Biosystems, Gaithersburg, MD, USA) according to the supplier's instructions, using the Nucleofector program X-005 (70% of transfection efficiency). Cells were suspended in nucleofection solution (10$^6$ cells/100 μl solution, Amaxa Biosystems) and mixed with 2 μg of the serum response element (SRE)-luciferase reporter plasmid (Cignal Reporter Assay Kits, SA Bioscence Corporation, BIOMOL GmbH, Hamburg http://www.sabiosciences.com/reporter_assay_product/HTML/CCS-010L.html), which allows for monitoring the activation of the MAPK/ERK signaling pathway. After 24 h, cells were exposed to the appropriate compounds or vehicles for 18 h. Quantification of Firefly and Renilla luciferase activity was performed using the Dual-Luciferase Reporter Assay System (Promega, France), according to the manufacturer's instructions. Renilla luciferase was used for internal normalization of Firefly activity values.

## Lentiviral transduction of GICs

The pCDH-CMV-MCS-EF1-puro HIV-based lentiviral vector (Systems Bioscience, USA) construct contains an ubiquitin promoter driving the expression of a luciferase-eGFP fusion product (Creusot *et al*, 2008). The luciferase gene is the Luc2 (pgl4) version (Promega, USA). The eGFP portion derives from the pIRES2-eGFP plasmid (Becton Dickinson, USA). Short hairpin RNA for PKCδ was from GE Dharmacon (GIPZ Lentiviral PKCD shRNA: E3-V3LHS_336853 and F10-V3LHS_637622). Lentiviral production and concentration were accomplished using standard protocols (Nitta *et al*, 2015). GICs were transduced for 12 h at 37°C, 5% CO$_2$, with lentivirus containing 6 μg/ml polybrene. After 24 h, cells were washed repeatedly to remove extracellular lentivirus. Cell sorting of eGFP-positive GICs was performed on a BD FACS ARIA (Becton Dickinson, USA).

## Intracranial xenografts

Freshly resected glioblastoma samples were obtained from the Department of Neurosurgery at Stanford University under approved institutional review board guidelines (IRB No. 18672) and dissociated using collagenase IV (1 mg/ml) and DNase I (250 units/ml) in HEPES-buffered HBSS and treated with ACK/RBC lysis buffer

(0.15 M NH$_4$Cl, 1.0 mM KHCO$_3$, and 0.1 mM Na2-EDTA). Enrichment in tumor-initiating cells was achieved by sorting EGFR$^+$/CD133$^+$ cells after immunolabeling of the primary cell suspension with anti-CD133-APC and anti-EGFR-Alexa 488 using a BD FACS ARIA (Becton Dickinson, USA). The cells were then plated for neurosphere formation, as previously described (Emlet et al, 2014). Passage 1 neurospheres were dissociated and transduced with a GFP-luciferase-encoding lentivirus, as described above. After secondary neurosphere formation, eGFP-positive cells were double-sorted to obtain a pure population of eGFP-positive neurosphere cells and 100,000 cells were injected stereotaxically into the striatum of anesthetized 6- to 8-week-old NOD scid gamma (NSG) mice, using the following coordinates: 2 mm posterior to the bregma, 2 mm lateral to the midline, and 3–4 mm deep with respect to the surface of the skull. For immunocompetent model, GL261 cells were injected into C57Bl/6 mice. Luminescent imaging was performed 90 days (for GBM44) and 240 days (for GBM5) after injection on an IVIS Spectrum (Caliper Life Science) and quantified using Living Image 4.0 software. D-luciferin (firefly) potassium salt solution (Biosynth) was prepared (16 mg/ml) and injected intraperitoneally (0.139 g luciferin per kg body weight). Total flux (photons per second) values were obtained by imaging mice until peak radiance was achieved and quantified with Live Image 4.0 software. Eleven mice were grafted for each group of treatment. Once tumor masses were detected, mice were randomized in two groups and prazosin (1.5 mg/kg or 0.15 mg/kg) or DMSO were delivered intraperitoneally twice a week for 45 days. Bioluminescent imaging was repeated at the end of the treatment and analyzed in a blind manner. At least two mice were euthanized at the end of the treatment for further histological examination. The remaining mice were used to assay survival (at least $n = 8$ per group of treatment). For histological analysis, the brains were kept in 4% paraformaldehyde at 4°C for 24 h, followed by 70% ethanol at room temperature for 24 h. Brains were then embedded in paraffin for 3 h at 67°C. Coronal sections (5 μm thick) were stained with hematoxylin and eosin, and images were acquired (Eclipse E800, Nikon, USA). All animal maintenance, handling, surveillance, and experimentation were performed in accordance with and approval from the Stanford University Administrative Panel on Laboratory Animal Care (Protocol 26548).

### Flow cytometry

Mice were euthanized, and brain tumors were dissociated to single cells and stained with anti-CD133-APC or anti-CD15 or anti-EGFR (Miltenyi) or Annexin V-PE (BioLegend) and DAPI. Tumor cells were gated based on GFP expression, and mouse cells were gated out using a lineage mixture of Pacific blue-conjugated H2k$^b$, H2k$^d$ Ter119, CD31, and CD45 antibodies. Flow cytometric analysis and cell sorting were performed on a BD FACS ARIA II (Becton Dickinson).

### Statistical analysis

Viability, proliferation, luciferase, and cell death experiments were performed in biological quadruplicates, in three independent experiments. Western blots, immunofluorescence, and flow cytometry images are representative from three independent experiments.

**The paper explained**

**Problem**

Glioblastoma is the most common and aggressive form of primary malignant brain tumor and has a very poor prognosis, mainly due to the glioblastoma-initiating cells (GICs). Since the current standard of care for glioblastoma is unable to eliminate the GICs, it is imperative to identify GIC-targeted therapies, especially those without major deleterious side effects on normal cells.

**Results**

Here, we found that prazosin, an α-adrenergic receptor antagonist, induces selective apoptosis of patient-derived GICs and differentiated GICs. We show that prazosin treatment inhibits glioblastoma growth in orthotopic xenograft mouse models and increases mice survival, with no toxicity. We provide evidence that the prazosin-induced GIC apoptosis occurs through PKCδ-dependent inhibition of AKT pathway, independent from adrenergic receptors.

**Impact**

Our findings identify prazosin as a potent inhibitor of glioblastoma growth. Prazosin is an FDA-approved drug with a record of over 40 years of safe and effective clinical use. Prazosin-induced GIC apoptosis associated with the well-established and documented use of this drug in clinical settings opens a promising possibility for its use in glioblastoma treatment.

Results are presented as mean $\pm$ SD unless otherwise presented. Results were analyzed using nonparametric two-tailed Mann–Whitney test to compare two groups. The level of significance was set at $P < 0.05$, as compared with the control group. Kaplan–Meier survival curves were compared using the log-rank (Mantel–Cox) test. Statistical analyses were carried out with Prism 6.0 software (GraphPad).

**Expanded View** for this article is available online.

## Acknowledgements

The authors thank Amélia Dias-Morais and Nadine Léonard for excellent technical assistance, Renaud Wagner and Fatima Alkhalfioui for help in binding experiments, and Salwa Sayd, Cécile Thirant, and Ashwin Narayanan for their constant support. This work was supported by Ligue Nationale Contre le Cancer (HC, MPJ, Equipe Labelisée), CAPES/COFECUB grant Me 757/12 (SAK and LGD PhD fellowships), CAPES (PLCC PhD fellowship), and CNPq (SAK and LGD PhD fellowships, SLC research fellowship), PEW Latin American Fellowship (SAK postdoctoral fellowship), Price Family Charitable Fund, Center for Children's Brain Tumors (SSM, and SHC), St Baldrick's Foundation, American Brain Tumor Foundation (SHC). SHC is the Tashia and John Morgridge Faculty Scholar and Ty Louis Campbell Foundation St. Baldrick's Scholar. SSM is the Seibel Stem Cell Institute Scholar. La ligue contre le cancer Comité du Haut-Rhin and Inserm, the French Ministère de l'enseignement supérieur et de la recherche (fellowship to MF), and the French government managed by "Agence Nationale de la Recherche" under "Programme d'investissement d'avenir" (LABEX ANR-10-LABX-0034_Medalis).

## Author contributions

JH and HC participated in conception and design. SAK, SG, RN, JH, MCK, LGD, and JC participated in development of methodology. SAK, SLC, MF, MZ, PV, BD, PLCC, and EEH participated in acquisition of data. SAK, MPJ, JH, HC, MCK, PV,

BD, and SSM participated in analysis and interpretation of data.MPJ wrote the manuscript. All authors equally contributed to review and/or revision of the manuscript.MCK, HC, MPJ, VMN, BD, and SHC provided administrative, technical, or material support. HC, MPJ, SHC, JH, MCK, and VMN supervised the study.

## Conflict of interest

The authors declare that they have no conflict of interest.

## For more information

http://www.ibps.upmc.fr/fr/Recherche/umr-8246/plasticite-gliale

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
