## [Review Process File · EMBO Molecular Medicine]

The anti-hypertensive drug prazosin inhibits glioblastoma growth via the PKC δ -dependent inhibition of the AKT pathway

Suzana Assad Kahn, Silvia Lima Costa, Sharareh Gholamin, Ryan T. Nitta, Luiz Gustavo Dubois, Marie Fève, Maria Zeniou, Paulo Lucas Cerqueira Coelho, Elias El-Habr, Josette Cadusseau, Pascale Varlet, Siddhartha S. Mitra, Bertrand Devaux, Marie-Claude Kilhoffer, Samuel H. Cheshier, Vivaldo Moura-Neto, Jacques Haiech, Marie-Pierre Junier, Hervé Chneiweiss

Corresponding author: Hervé Chneiweiss, Pierre and Marie Curie University

Review timeline:

Submission date:	07 May 2015
Editorial Decision:	15 June 2015
Revision received:	15 December 2016
Editorial Decision:	01 February 2016
Revision received:	17 February 2016
Accepted:	19 February 2016

Transaction Report:

Editor: Roberto Buccione

1st Editorial Decision

15 June 2015

Thank you for the submission of your manuscript to EMBO Molecular Medicine. We are sorry that it has taken longer than usual to get back to you on your manuscript. In this case we experienced some difficulties in securing three appropriate reviewers and then obtaining their evaluations in a timely manner and also we needed to discuss your manuscript further.

As you will see three Reviewers are positive, but do raise many issues, quite a few of which are fundamental and overlapping. Although I will not dwell into much detail, I would like to highlight the main points.

You will see that there are a few recurring themes. Among these, the need to better substantiate the contention that Prazosin acts via PKC δ , the fact that the levels of Prazosin (and approved by the FDA) that exert behavioural effects are lower than those required for its anti-cancer activity and might not be attainable in vivo and that more evidence is needed to support the suggestion that the GICs are the main target.

The Reviewers also suggest other specific items for your action. Of note you will see that Reviewer 3 asks for more details on the in vivo experiments and statistical treatment. We fully agree and

indeed inform you that that EMBO Molecular Medicine now requires a complete author checklist (<http://embomolmed.embopress.org/authorguide#editorial3>) to be submitted with all revised manuscripts. Provision of the author checklist is mandatory at revision stage; The checklist is designed to enhance and standardize reporting of key information in research papers and to support reanalysis and repetition of experiments by the community. The list covers key information for figure panels and captions and focuses on statistics, the reporting of reagents, animal models and human subject-derived data, as well as guidance to optimise data accessibility.

In conclusion, while publication of the paper cannot be considered at this stage, given the potential interest of your findings and after internal discussion, we have decided to give you the opportunity to address the above concerns. We are thus prepared to consider a substantially revised submission, with the understanding that the Reviewers' concerns must be addressed in toto, with additional experimental data where appropriate and that acceptance of the manuscript will entail a second round of review.

I appreciate that if you do not have the required data available at least in part, to address the above, this might entail a significant amount of time, additional work and experimentation and might be technically challenging, I would therefore understand if you chose to rather seek publication elsewhere at this stage. Should you do so, we would welcome a message to this effect.

Please note that it is EMBO Molecular Medicine policy to allow a single round of revision only and that, therefore, acceptance or rejection of the manuscript will depend on the completeness of your responses included in the next, final version of the manuscript.

As you know, EMBO Molecular Medicine has a "scooping protection" policy, whereby similar findings that are published by others during review or revision are not a criterion for rejection. However, I do ask you to get in touch with us after three months if you have not completed your revision, to update us on the status. Please also contact us as soon as possible if similar work is published elsewhere.

***** Reviewer's comments *****

Referee #1 (Novelty/Model system Comments for Author):

The technical quality of the study may be improved by giving more insight on the signalling pathways modulated by Prazosin: Manipulating PKC-delta activation molecularly (and not only by Rottlerin) would largely improve technical quality.

The use of Prazosin is a novel approach for gliomas, but has been discussed for other / peripheral tumours - hence novelty is not my highest rating.

The medical impact may nevertheless be considered as high - given that Prazosin enters the brain in sufficient quantity to mediate the desired effect.

There are concerns regarding the model system as it is currently unclear why the authors selected a certain set of tumour cells.

Referee #1 (Remarks):

The study entitled "Prazosin induces apoptosis in glioblastoma through PKC-delta-dependent inhibition of AKT pathway" explores the anti-tumour effects of the clinically approved (non-selective) alpha-adrenergic antagonist Prazosin on malignant brain tumours (GBM). Here, Kahn et al. explore different alpha-adrenergic antagonists and show that only Prazosin reduces GBM cell

viability - albeit at remarkably high doses (more than 5 micromolar). This anti-tumour effect is interesting as it seems to be GBM cell-specific, since some human neural stem cell lines (hNSC) are largely unaffected by the drug while different human primary GBM lines are sensitive. Furthermore the authors claim that Prazosin targets a highly aggressive subpopulation of GBM cells, so-called glioma initiating cells (GICs) and suppresses GIC-mediated tumourigenesis in vitro and in vivo. Consistently, a survival study using GBM implanted mice showed that systemic Prazosin application could somewhat prolong survival (although the therapeutic effect was not large). Prazosin may induce GBM cell apoptosis by an off-target effect of the drug. The authors provide some evidence that Prazosin modulates Protein-Kinase-C-delta (PKC-d) activity which can affect AKT signalling and indirectly promote caspase-3 activation. Caspase-3 may cleave PKC-d and a lower molecular weight form of PKC-d may accelerate GBM death in a feed-forward loop. Overall, this study suggests to use a clinically available drug to more efficiently treat GBM (which currently have a very poor prognosis). This is potentially interesting, and (to my opinion) deserves further exploration. However, some issues concerning the studied GBM cells, the control cells (hNSC), the penetrance of the drug into the brain and the suggested signalling mechanisms remain.

Major points:

1. Kahn et al. use 4 different human primary GBM cultures for their study (TG1, TG10, TG16, TG19). It is unclear how and why these four lines were selected for the presented experiments. Some of these lines (TG16 and TG19) are derived from giant cell GBM (which is a rare GBM variant), while other lines (previously used by the authors) were not used for the present study. Although the authors claim that all lines were evaluated previously I could not find information on line TG10. Hence, a broader experimental set-up using more and better defined GBM cultures is necessary to substantiate the findings.
2. The authors claim that Prazosin has anti-tumour effects against GICs. However, in none of the studies cited any conclusive evidence is provided that TG1, TG10, TG16, TG19 are enriched in GICs. Hence, the data support a role for Prazosin in GBM cell-death induction, but experiments specifically addressing the glioma-initiating capacity of a subpopulation of GBM cells are not provided.
3. The experiments using hNSC are of potential interest as these cells may be used to show the specificity of the drug (against neoplastic cells). However, it would be more interesting to first induce the differentiation of hNSCs (thereby generating neurons astrocytes and a small population of oligodendrocytes) and then to test the drug on these relevant cells. Furthermore some note of caution should be added to the text, as all these hNSCs are embryonic and may be different from adult human brain cells.
4. Prazosin is efficient only at very high doses and currently it is not clear if these high drug concentrations can be reached in the brain. Some reports show behavioural effects mediated by Prazosin - but these could be caused by much lower drug concentrations. Therefore, it is vital to extend the in vivo experimentation and also to provide more insight into the specific mechanisms of drug-action. So far, the authors have only addressed the issue of Prazosin-mediated PKC-d activation by blocking PKC-d with Rottlerin. This is insufficient since Rottlerin has many off-target effects. Knock-down (or Crispr/Cas9 induced knock-out) of PKC-d are necessary to show the specific mechanism. Overexpression of a knock-down resistant PKC-d (rescue) and a cleavage-resistant PKC-d will give much better insight into the signalling pathway. These PKC-d manipulated cells should then be used in vivo to address the issue of Prazosin mediated therapeutic effects again. This will show if sufficient drug enters the brain to induce the described (PKC-d specific) cell-death pathway. The current data (with survival experiments showing some therapeutic effect in vivo) may also be due to other effects reported for Prazosin like e.g. reduced intratumoral angiogenesis.

Minor point:

There is already one study presenting effects of Prazosin against tumour stem cells (which was not cited in the present text): BMC Cancer. 2014 Feb 14;14:90. Protein kinase C- inactivation inhibits the proliferation and survival of cancer stem cells in culture and in vivo. Chen Z, Forman LW,

Williams RM, Faller DV.

Referee #2 (Remarks):

Targeting glioma initiating cells (GICs) is a potentially exciting approach for glioma therapy and appropriate for EMM. The current manuscript demonstrates an impressive effect of prazosin (PRZ) both in culture and in vivo. There are some studies on potential mechanisms, which are not as strong. I would suggest that the authors should consider a number of areas to strengthen the studies.

Major concerns:

1. The effects of PRZ are surprisingly stronger than other drugs of the same class. Can the authors explain why? In an revised manuscript experiments, these other agents may be excellent for controls.
2. Can the authors comment on AR, PKC, etc. expression in the models and possible links to sensitivity?
3. The in vitro studies would benefit from in vitro limiting dilution studies.
4. What are the effects of PRZ on differentiated tumor cells? Are the putative molecular targets differentially expressed or activated between GICs, differentiated tumor cells, and normal brain cells?
5. The in vivo treated tumors would benefit from more direct analysis to investigate the cause of effects. Is apoptosis occurring in vivo? Specifically in GICs? Can the authors show any functional change in GICs after treatment?
6. A second in vivo model would be valuable.
7. The most challenging parts are the last two figures. The effects are modest and the rescue effects are modest. As this is a molecular medicine journal, I would suggest more development (more lines, better rescue studies, more phenotype). Other GIC targets (Ephs, NO synthetase, etc.) have been proposed as possible targets for PRZ and other related drugs.

Referee #3 (Novelty/Model system Comments for Author):

The prospect of adopting an FDA-approved drug that penetrates BBB for GBM is very appealing. However, the drug prazosin identified in this study, and the off-target mechanism of its action leading to reduced viability of glioma initiating cells (GICs), requires further cautious examination in vitro and in vivo. The concluding key role of PKC in prazosin-induced GIC death is largely based on experiments with rottlerin, a molecule that is not selective to PKC. Furthermore, descriptions of the experiments performed in this study are not always sufficient to evaluate the technical quality of the work.

Specific comments and questions:

1. While prazosin reduces GIC viability, a more detailed analysis of its effects is lacking. The cell death was accompanied by inhibition of GIC proliferation of the surviving cells, but the relationship between proliferation and cell death have not been studied. Does prazosin affect sphere-forming capacity and/or cell cycle? If there is a sub-population of resistant cells, what properties does it have? If the effect is mostly pro-apoptotic, further detailed analysis of the cell death inducing signaling should be carried out. Are additional caspases activated?

2. The key experiments are based on rottlerin, whose role as a specific PKC inhibitor is highly questionable. It inhibits various kinases including GSK3, and uncouples mitochondrial oxidative phosphorylation. To claim the central role of PKC in prazosin-induced apoptosis, additional and more specific inhibitors (e.g. siRNAs) should be explored.

3. Figs. G-J demonstrate that PRZ treatment results practically in the disappearance of tumors, however, why is survival not substantially prolonged in this case? Are these figures representative? More detailed immunohistological analysis of tumor sections (e.g. for Ki67, TUNEL) would be informative.

4. It is not stated how many animals were included in the in vivo experiments. From Fig. 2B-E, it seems that there were only 4 mice per group; if this is the case, this number should be statistically justified. Further, the Kaplan Meyer survival for one xenograft model is shown only. What were the results from the second model studied?

5. Prazosin reduced GICs viability with an EC50 value of 7.88 nM, several orders of magnitude above the nanomolar concentrations at which it acts on α_1 -ARs. The dose of 5 mg/kg administered to tumor-bearing mice is also significantly higher than the FDA approved. It would be important to test if lower doses, in a range approved by the FDA, will effectively inhibit tumor growth.

6. It would be also important to test the effects of the drug in immunocompetent GBM models.

Minor comments:

The figure legends are not always sufficiently explanatory. For example, what's shown in Fig. 2 G/J? Are those two different xenograft models or two animals per group representing the same model?

Fig. 2F: it should be labeled when the treatment was initiated.

Dear Dr Buccione,

Please find enclosed our revised manuscript entitled “Prazosin induces apoptosis in glioblastoma through PKC δ -dependent inhibition of AKT pathway”, EMM-2015-05421. This revised version contains the results of the novel experiments suggested by the reviewers to address their concerns.

Please, note that we added in the point-by-point answer to the referees, patient data that must remain confidential.

We took into account your statement that “*Reviewers are positive, but do raise many issues, quite a few of which are fundamental and overlapping.. You will see that are a few recurring themes*”, by grouping the referees concerns relating to a similar theme when it was adapted, and providing a single and comprehensive response. We do hope that this presentation will ease the reviewing process, and that the large amount of new data, particularly *in vivo*, will satisfy the reviewers. Anyhow, we are grateful to the reviewers for their thoughtful comments that we believe clearly helped to strengthen our conclusion that Prazosin acts on Glioma Initiating cells via PKC delta.

We hope that the corrections made will satisfactorily address their questions.

We thank you for your consideration and your support,

Best regards

Hervé Chneiweiss and Marie-Pierre Junier

Answers to Reviewer's comments

Referee #1, point 4 of the Novelty/Model system Comments for Author: “There are concerns regarding the model system as it is currently unclear why the authors selected a certain set of tumour cells.”

and Major point 1: “1. Kahn et al. use 4 different human primary GBM cultures for their study (TG1, TG10, TG16, TG19). It is unclear how and why these four lines were selected for the presented experiments. Some of these lines (TG16 and TG19) are derived from giant cell GBM (which is a rare GBM variant), while other lines (previously used by the authors) were not used for the present study. Although the authors claim that all lines were evaluated previously I could not find information on line TG10. Hence, a broader experimental set-up using more and better defined GBM cultures is necessary to substantiate the findings.”

Considering the fact that Glioma Initiating Cells were previously demonstrated to be highly resistant to current treatments, we chose GICs that resisted to up to 1mM temozolomide. This was the case for TG1, TG10 and TG16 cells used in the present study (table 6 of Patru et al. BMC Cancer, 2010). We believe that these cells are the targets of choice for any novel treatment aiming to stop glioblastomas growth. TG19 cells, which were subsequently derived from another glioblastoma exhibit the same properties. The characterization of the TG10 line was reported in Patru et al. BMC Cancer 2010 (pages 6 and 7 of the article), and was isolated from a giant cell glioblastoma. TG16 was also reported in Patru et al. BMC Cancer 2010 and was isolated from a classical glioblastoma. Thanks to the reviewer’s remark, we discover that a mistake appeared in the table 1 of Silvestre et al. 2011 since the line for TG10 was duplicated. We apologize for this error. We provide as attached files the original diagnosis of the patients. TG10 and TG16 were chosen because they present a classical loss-of-function mutation of TP53 whereas TG1 and TG19 express a wild type form of TP53, like the TG18 published in Silvestre et al. Stem Cells 2011. These two kinds of cells were used to demonstrate that prazosin effects are not dependent on TP53. To take into account the referee demand, we now present in the revised version the results obtained with two other cell lines, GBM5 and GBM44 (method of characterization described in Emler et al. Cancer Research 2014), both derived from surgical resections of “classical” primary glioblastomas. We believe this choice reinforces the coherence of the corpus of data since GBM5 and GBM44 were used for the *in vivo* experiments, and further strengthens our demonstration, since GBM5 and GBM44 were obtained in a different country, United States, at Stanford University. To remain on the same grounds, we now present two human NSC cell lines characterized in Paris (and previously published in Thirant et al. PLoS One 2011) and two others obtained in the US (NSC5031 and NSC8853). Accordingly, Fig 1 was modified with a new panel C presenting the results obtained with these different cells. Of note, the results remain the same, namely that prazosin triggers a significant cell death in GICs from 5 to

30 μ M whereas NSC are only slightly affected and only above 10 μ M prazosin. The corresponding texts in the results (page 3), the Materials and Methods (page 8, last paragraph), and the figure legends (page 18) sections were modified accordingly.

Referee #1, Major point 2: “2. The authors claim that Prazosin has anti-tumour effects against GICs. However, in none of the studies cited any conclusive evidence is provided that TG1, TG10, TG16, TG19 are enriched in GICs. Hence, the data support a role for Prazosin in GBM cell-death induction, but experiments specifically addressing the glioma-initiating capacity of a subpopulation of GBM cells are not provided.”
and Referee #2, Major point 3: “3. The in vitro studies would benefit from in vitro limiting dilution studies.

We now provide three additional sets of evidence supporting that prazosin affects GICs.

1- We first followed the recommendation of reviewer 2 and performed extreme limiting dilution assays (ELDA, <http://bioinf.wehi.edu.au/software/elda/>). We observed that prazosin induced a drastic reduction in the number of sphere-forming cells. These results are now presented in Fig 1 panel D and Expanded View Fig 1 for TG1 and GBM44 cells, respectively. Texts were modified accordingly in the following sections: Results page 3, lines 31-34; Materials and Methods page 9; Legends pages 18 and 21.

2- The second test was to sort the cells according to their expression of EGFR, a marker of malignancy, and of CD133 and CD15, frequently used as GIC markers. The results presented in Fig 1E show that prazosin reduces the viability of every subtype including EGFR+/CD133+/CD15+ cells.

These results are now described as follows (results section, page 3, lines 31-37):

“Extreme limiting dilution assay (ELDA) was used to further evaluate the targeting of GICs by prazosin. Frequency of sphere-forming cells, a surrogate property of GICs (Flavahan et al, 2013) was drastically reduced by prazosin, dropping from 1/3.88 to 1/248 for TG1 ($p = 1.13 \cdot 10^{-10}$) and from 1/6.32 to 1/31 for GBM44 ($p = 0.0331$) (Fig 1D and Fig EV1). In addition, we sorted the GIC according to their expression of EGFR, a marker of malignancy, and of CD133 and CD15, frequently used as GIC markers (Son et al, 2009); Mazzoleni et al, 2010; Emler et al, 2014). Prazosin inhibited the survival of every population subtype, including EGFR+/CD133+/CD15+ cells (Fig 1E).”

3- Finally, we also evaluated glioma cells *in vivo* in two ways. First, we analyzed the numbers of CD133+ cells in treated versus untreated mice. We observed a significant reduction of the CD133+ population following prazosin treatment (Fig. 2E). Then we performed secondary xenografts, using cells obtained from tumors developed in mice treated or not with prazosin. The results showed that cells isolated from primary tumors of prazosin-treated mice exhibited a reduction in their tumor-initiation property, a core characteristic of GICs (Fig. 2F), suggesting a drastic reduction in GICs number.

These results are now described as follows (results section, Page 4, lines 13-21:

“Flow cytometry analysis of GFP-positive tumor cells showed a significant decrease in human CD133-positive cells in prazosin-treated mice, suggesting removal of GICs along with the non-GICs (Fig 2E). To further demonstrate that prazosin affects GICs, we evaluated its effects on a major property of cancer stem cells, tumor initiation. GFP-positive tumor cells from primary tumors were isolated (see Materials and Methods section) and reinjected into new groups of mice (Fig 2F). All mice grafted with glioblastoma cells isolated from vehicle-treated mice developed tumors (8/8 cases, Fig 2F). On the other hand, only 4/8 mice injected with glioblastoma cells isolated from prazosin-treated mice developed tumors (Fig 2F). Moreover, mice injected with glioblastoma cells isolated from prazosin-treated mice presented a statistically significant survival benefit ($p = 0.0047$) (Fig 2F).”

Referee #1, Major point 3: “3. The experiments using hNSC are of potential interest as these cells may be used to show the specificity of the drug (against neoplastic cells). However, it would be more interesting to first induce the differentiation of hNSCs (thereby generating neurons astrocytes and a small population of oligodendrocytes) and then to test the drug on these relevant cells. Furthermore some note of caution should be added to the text, as all these hNSCs are embryonic and may be different from adult human brain cells.”

and Referee #2, Major point 4: “What are the effects of PRZ on differentiated tumor cells? Are the putative molecular targets differentially expressed or activated between GICs, differentiated tumor cells, and normal brain cells?”

We thank the reviewers for these suggestions that allowed us to show that prazosin acts on both GICs and their progenies. Using ad hoc media NSC or GIC were differentiated along the astroglial, oligodendroglial and neuronal lineages as demonstrated by increased expression of GFAP, O4 and β 3-Tubulin respectively (Fig. 1F). Differentiated tumor cells were highly affected by prazosin, whereas neurons, astrocytes and oligodendrocytes generated from NSC were minimally affected (Fig. 1G). Furthermore these results were validated *in vivo* since, as illustrated in Fig 4D, prazosin did not induce apoptosis in GFP-negative cells (i.e. non-tumor stromal cells of the adult mouse brain). To follow the reviewer's suggestion, we also added in the text a note of caution about the embryonic nature of hNSC. Of note, the differentiation did not modify the levels of PKC δ expression in tumor cells (see Expanded View Fig 4A).

The Results and the Discussion sections have been modified as follows:

-Results, page 3, last 2 lines and page 4 lines 1-2:

"To further evaluate whether the effectiveness of prazosin is influenced by the stem and/or differentiated state of the cells, NSCs and GICs were differentiated along the astroglial, oligodendroglial and neuronal lineages (Fig 1F). Prazosin inhibited also the survival of differentiated glioblastoma cells whereas minimally affecting differentiated NSCs (Fig 1G)."

-Discussion, Page 8 lines 4-7:

"An effect of prazosin on adult human neural cells cannot be excluded since we used human NSCs of embryonic origin, although no deleterious effect of prazosin was observed (see Fig 2D, Fig 3D, and Fig 4D) or has been reported so far in mouse brain following administration of doses akin to the ones we used."

Referee #1, Major point 4: "Prazosin is efficient only at very high doses and currently it is not clear if these high drug concentrations can be reached in the brain. Some reports show behavioural effects mediated by Prazosin - but these could be caused by much lower drug concentrations. Therefore, it is vital to extent the in vivo experimentation and also to provide more insight into the specific mechanisms of drug-action. So far, the authors have only addressed the issue of Prazosin-mediated PKC-d activation by blocking PKC-d with Rottlerin. This is insufficient since Rottlerin has many off-taget effects. Knock-down (or Crispr/Cas9 induced knock-out) of PKC-d are necessary to show the specific mechanism. Knock-down (or Crispr/Cas9 induced knock-out) of PKC-d are necessary to show the specific mechanism. Overexpression of a knock-down resistant PKC-d (rescue) and a cleavage-resistant PKC-d will give much better insight into the signalling pathway. These PKC-d manipulated cells should then be used in vivo to address the issue of Prazosin mediated therapeutic effects again. This will show if sufficient drug enters the brain to induce the described (PKC-d specific) cell-death pathway. The current data (with survival experiments showing some therapeutic effect in vivo) may also be due to other effects reported for Prazosin like e.g. reduced intratumoral angiogenesis."

and Referee #2 point 7:" The most challenging parts are the last two figures. The effects are modest and the rescue effects are modest. As this is a molecular medicine journal, I would suggest more development (more lines, better rescue studies, more phenotype)."

and Referee#3 point 2: The key experiments are based on rottlerin, whose role as a specific PKCdelta; inhibitor is highly questionable. It inhibits various kinases including GSK3beta;; and uncouples mitochondrial oxidative phosphorylation. To claim the central role of PKCdelta; in prazorin- induced apoptosis, additional and more specific inhibitors (e.g. siRNAs) should be explored.

Regarding the penetration of prazosin within the brain and the *in vivo* concentrations used:

To document the penetration of prazosin within the tumor *in vivo*, we performed intra-peritoneal injections of the green-fluorescent derivative of prazosin, BODIPY FL prazosin. The results presented in Fig 3E show that prazosin reaches the tumor *in vivo* within 1 to 2 hours post-injection.

Description of this result is provided page 4 lines 30-32:

"Finally, using this glioblastoma model coupled with intra-peritoneal injections of the green-fluorescent derivative of prazosin, BODIPY FL prazosin, we observed a marked accumulation of prazosin in the tumor within two hours post-treatment (Fig 3E)."

In the original submission, a typing error led to mention an erroneous dose of 5mg/kg of prazosin administrated *in vivo* instead of the 1.5mg/kg dose, which was always used. We apologize for this error, which has been corrected in the revised manuscript. We also performed novel *in vivo* experiments using 0.15 mg/kg of

prazosin, a dose compatible with the human daily regimen for treatment of hypertension. This lower dose resulted in a significant inhibitory effect on tumor growth associated with a survival benefit as shown in Fig 2G.

The corresponding results and discussion sections have been modified as follows:

-Results, page 4, lines 22-25:

“We also tested lower doses of prazosin (0.15mg/kg instead of 1.5mg/kg) compatible with the human daily regimen for treatment of hypertension (see Discussion section). The lower dose of prazosin also induced a significant reduction in tumor growth and increased survival of glioblastoma-bearing mice (Fig 2G).”

-Discussion, page 8, lines 15-23:

“The current use of prazosin hydrochloride in humans as an oral prescription for hypertension is, according to the FDA recommendation, a total daily dose of 20 mg that may be further increased up to 40 mg given in divided doses. Bioavailability studies have demonstrated peak levels of approximately 65% of the drug in solution (<http://www.fda.gov/Safety/MedWatch/SafetyInformation/Safety-RelatedDrugLabelingChanges/ucm155128.htm>).

The daily dosage already approved is therefore likely to result in a bioavailability of prazosin in the 5-10 μ M range, shown here to effectively induce GIC apoptosis *in vitro*. Such a potential use as adjuvant therapy in humans is further supported by our observations that low doses of prazosin (0.15mg/kg should be extrapolated to 9mg for an adult of 60kg) also significantly inhibited tumor growth *in vivo*.”

Regarding prazosin-mediated PKC δ activation. A great part of the additional data provided in this revised manuscript is dedicated to answer these reviewer’s requests. We used two novel additional means to evaluate the role of PKC δ in prazosin-induced glioblastoma cell death:

1- δ V1.1, a peptide that specifically opposes PKC δ mobilization. First described by the group of Daria Mochly-Rosen, a recognized expert of PKCs (Chen et al., PNAS 2001), it has been subsequently used in several other studies from this team as well as others..

2- shRNA targeting PKC δ (see its effectiveness in Expanded View Fig 4B).

The novel results are now combined with the previously reported ones in the novel Fig 5. δ V1.1 significantly prevented cell death induced by prazosin (Fig 5F), as did shPKC δ (Fig 5G). Accordingly δ V1.1 counteracted the inhibitory effects of prazosin on AKT phosphorylation (Fig 5L).

The text was implemented accordingly

Page 6, lines 3-6:

“In addition to rottlerin, we used the peptide δ V1.1 that specifically opposes PKC δ mobilization [Chen, 2001], and silenced the expression of PKC δ using shRNA (Fig EV4B). Rottlerin (Fig 5E), δ V1.1 (Fig 5F) and PKC δ shRNA (Fig 5G) significantly rescued GICs from prazosin-induced glioblastoma cell death.”

Page 6 lines 16-18

“We then exposed GICs to prazosin in the presence or absence of rottlerin (Fig 5K) or δ V1.1 (Fig 5L), and assessed the levels of P-AKT. Rottlerin as well as δ V1.1 counteracted prazosin-induced inhibition of AKT phosphorylation (Fig 5K-L).”

We also used cells transduced with PKC δ shRNA *in vivo*. The results now presented in the novel Fig 6 indicate that PKC δ mediates prazosin-induced glioblastoma cell death also *in vivo*.

The text was modified accordingly Page 6 lines 18-25:

“The role of PKC δ was further investigated *in vivo*. Silencing the expression of PKC δ with a shRNA (Fig EV4B) resulted in a slower growth of tumors (Fig 6B, Prior-PRZ images). This result suggests that PKC δ expression is necessary for GIC tumor growth *in vivo*, and is coherent with the reported inhibitory effects of PKC δ shRNA on the growth of xenografts of prostate, pancreas and breast cancer stem cells (Chen et al, 2014). We then analyzed the effect of prazosin in PKC δ -silenced tumors *in vivo*. As expected, prazosin inhibited the growth of control (shScramble) tumors (Fig 6B). On the other hand, PKC δ -silenced tumors were no longer responsive to prazosin treatment (Fig 6B-D), further confirming the involvement of PKC δ in prazosin-induced glioblastoma cell death.”

Regarding an eventual effect of prazosin on intra-tumoral angiogenesis:

We evaluated the density of vessels in xenografted mice brain treated or not with prazosin and found no difference, suggesting that effects of prazosin most likely are not directly on angiogenesis (Expanded View Fig 2C).

This result is now described on page 4, lines 12-14 as follows:

“Of note, tumors from vehicle and prazosin-treated mice presented similar blood vessels density suggesting that prazosin did not affect angiogenesis (Fig EV2C).”

Referee 1, Minor point: “There is already one study presenting effects of Prazosin against tumour stem cells (which was not cited in the present text): *BMC Cancer*. 2014 Feb 14;14:90. Protein kinase C-delta; inactivation inhibits the proliferation and survival of cancer stem cells in culture and in vivo. Chen Z, Forman LW, Williams RM, Faller DV.”

We thank the reviewer for calling our attention on this interesting article reporting on the effects of PKC δ inhibition on prostate, pancreas and breast cancer stem cells that we missed in our first submission. Interestingly, the authors observe that PKC δ inhibition, either using shRNA or different pharmacologic inhibitors, including rottlerin, result in a cell growth inhibition either tested as tumor-sphere or after xenograft, in good agreement with our observations of the effects of PKC δ shRNA on intracerebral xenografts of GICs. Of note, these authors do not use prazosin. We now cite this article page 6, lines 20-22 when we present the effects of PKC δ knock-down on xenograft growth.

Referee #2, Major concern 1: “1. The effects of PRZ are surprisingly stronger than other drugs of the same class. Can the authors explain why? In an revised manuscript experiments, these other agents may be excellent for controls.”

We agree with the reviewer that the selective effect of prazosin, and not the other adrenoreceptor antagonists, came as a matter of surprise; however this is not the only example among quinazoline-based alpha-adrenoreceptor antagonists. Lin et al. (*Neoplasia* 2007) reported that 10 μ M prazosin was more effective than 100 μ M Doxazosin to induce DNA damage in prostate cancer cells in vitro. Deciphering the bases of the differing actions of quinazolines is however beyond the aim of the present work. To acknowledge the fact that this question is still unresolved, we modified the discussion as follows (page 7, lines 11-14):

“The reason why prazosin is the most effective of the quinazolines tested on GICs remains to be determined. Interestingly, a preferential prazosin toxicity, among other quinazoline-based alpha-adrenoreceptor antagonists, has been reported on prostate cancer cells, 10 μ M prazosin being in this case more effective than 100 μ M doxazosin in inducing DNA damage (Lin et al, 2007).

To follow the reviewer’s recommendation, we used Terazosin as a negative control, since it does not alter GSC viability. The results show that Terazosin, which does not alter cell viability (Expanded View Fig 4C) has no effect on AKT activation (Fig. 5H). The text of the results was modified accordingly (Page 6, lines 8-10):

“Prazosin inhibited AKT phosphorylation in GICs as efficiently as LY294002, a specific PI3K inhibitor (Fig 5H, left panel). Terazosin, used as a control since it does not affect GICs survival (Fig EV4C), did not modify AKT phosphorylation (Fig 5H, right panel).”

Referee #2, Major concern 2: “2. Can the authors comment on AR, PKC, etc. expression in the models and possible links to sensitivity?”

As already indicated in the original submission, we think that GIC sensitivity to prazosin is due to their high expression of PKC δ in comparison to NSC. As novel data, we show in Fig 5I that prazosin does not affect the level of phosphorylation/activity of AKT in NSC, suggesting that a preliminary activation of PKC δ is mandatory for prazosin effects. We overexpressed PKC δ in NSC. We could verify the overexpression but unfortunately could not assay the cell response to prazosin since NSC-overexpressing PKC δ did not survive.

We have now discussed this point as follows (page 7, last line, page 8, lines 1-4):
“Although the signaling pathways sustaining NSCs and GICs maintenance differ in several ways, they both require a proper functioning of the PI3K/AKT pathway for their survival (Groszer et al, 2006; Yan et al, 2013). Accordingly, no change in AKT phosphorylation was observed in NSCs following prazosin treatment. This result, associated with the paucity of PKC δ levels in NSCs as compared to GICs, suggests that a preliminary activation of PKC δ is mandatory for prazosin to exert its pro-apoptotic action.”

Referee #2, Major concern 3: we addressed this point page 2 of this letter Page 3 of this letter together with answer to referee #1 point 2

Referee #2, Major concern 4: we addressed this point page 3 of this letter together with answer to referee #1 point 3

Referee #2, Major concern 5: “5. *The in vivo treated tumors would benefit from more direct analysis to investigate the cause of effects. Is apoptosis occurring in vivo? Specifically in GICs? Can the authors show any functional change in GICs after treatment?*”

To address *in vivo* apoptosis, we sorted cells from tumors treated or not with prazosin, and used DAPI and Annexin V staining to identify apoptotic cells. The results show that Prazosin induces apoptosis in glioblastoma cells *in vivo* (Fig. 4D). We also performed TUNEL analysis on tumor sections, and observed increased numbers of TUNEL+ cells (Expanded View Fig 2B). Technical issues having prevented the *in situ* co-staining of TUNEL with stem-like markers, we used secondary xenografts of cells sorted from primary tumors in control or prazosin-treated mice to document the targeting of GICs *in vivo*. As described above in our response to point 2 of referee #1 (page 2 of this letter), we observed that the growth of tumors initiated by cells sorted from prazosin-treated tumors is reduced as compared to controls (Fig 2G). These results indicate a drastic reduction of the number of tumor initiating cells in prazosin-treated tumors.

Referee #2, Major concern 6: “6. *A second in vivo model would be valuable.*”
and referee #3, point 6 : “6. It would be also important to test the effects of the drug in immunocompetent GBM models.”

We had already presented in the original submission two *in vivo* models performed with two distinct GIC lines. The second model might have escaped referee #2 attention because the survival data of the second model was missing. We now present a third *in vivo* model to answer referee #3 demand to use an immunocompetent model. The results obtained with implantation of the mouse glioblastoma-like cell line GL261 in C57/Bl6 mouse brain were similar to the ones obtained with xenografts of human GIC in immunodeficient mice. The results are presented in the novel Fig 3. The text of the results section was implemented accordingly in page 4 lines 23-26:

“To verify whether prazosin effects could also be observed in an immunocompetent syngeneic mouse model, we implanted the mouse glioblastoma-like cell line GL261, transduced with GFP-luciferase, in C57/Bl6 mouse brain. Prazosin induced GL261 cell death *in vitro* (Fig 3A), and significantly inhibited tumor growth *in vivo* (Fig 3B-D), an effect associated with a survival benefit (Fig 3C).”

Referee #2, Major concern 7: “*The most challenging parts are the last two figures. The effects are modest and the rescue effects are modest. As this is a molecular medicine journal, I would suggest more development (more lines, better rescue studies, more phenotype). Other GIC targets (Ephs, NO synthetase, etc.) have been proposed as possible targets for PRZ and other related drugs.*”

We addressed the first part of this point on the Page 3 of this letter together with the answer to referee # 1 point 4, and referee #3 point 2.

Regarding the other possible prazosin targets: as mentioned by the referee, other targets have been proposed for prazosin but most of them such as Ephs, HERG ligand, EGFR inhibition, etc are also targets for doxazosin and terazosin, which were poorly effective in our model. We now provide in addition the demonstration that Terazosin does not activate PKC δ (Fig 5H). We cannot exclude that other and complementary pathways are activated by prazosin and might be also activated by other quinazolines of the same family. However, our data show that GIC death is preferentially induced by prazosin because of its targeting of PKC δ .

Referee #3, Novelty/Model system Comments for Author: “.../... *Furthermore, descriptions of the experiments performed in this study are not always sufficient to evaluate the technical quality of the work.*”

We have now clarified the schemes of the *in vivo* protocols presented in the figures, and revised the Materials and Methods section.

Referee #3, Specific comments and questions 1: “1. While prazorin reduces GIC viability, a more detailed analysis of its effects is lacking. The cell death was accompanied by inhibition of GIC proliferation of the surviving cells, but the relationship between proliferation and cell death have not been studied. Does prazorin affect sphere-forming capacity and/or cell cycle? If there is a sub-population of resistant cells, what properties does it have? If the effect is mostly pro-apoptotic, further detailed analysis of the cell death inducing signaling should be carried out. Are additional caspases activated?”

To clarify these aspects and follow the reviewer’s suggestions, we added results not presented in the first submission, performed new experiments and reorganized the presentation of the results.

Extreme limiting dilution assays demonstrate that prazosin affects sphere forming capacity (Fig 1D and Expanded View Fig 1). We also analyzed DAPI and Annexin V staining of GICs, which were sorted according to their expression of the neural stem cell marker CD15 prior to be treated with 10 μ M prazosin. The results demonstrate that prazosin induces apoptosis of a majority of CD15⁺ and CD15⁻ cells (Expanded View Fig 2A). In addition, we performed secondary grafts to confirm that prazosin targets GICs *in vivo*. Please, see also our answer to referee #1 point 2, and referee #2 point 3, page 2 of this letter.

We also verified the behavior of GICs having survived to a first 72 h prazosin treatment. GICs were treated with prazosin for 72 h, the medium was then replaced with fresh medium and the cells allowed to recover for 2 weeks prior to be exposed to prazosin for 72 h again. The results presented in Expanded View Fig 2D show that a second prazosin treatment reduced the survival of these cells. These results are described as follows (page 3, first paragraph of the results, lines 26-28):

“In addition, we explored whether GICs having escaped a first 72 h prazosin-treatment were responsive to a second prazosin treatment. The results showed that GICs remained sensitive to 30 μ M prazosin (Fig EV2D).”

To clarify the presentation of the data concerning cell death, the new Fig 4 is devoted to the demonstration that (1) prazosin-induced glioblastoma cells death is through apoptosis (panels A-D) and (2) prazosin acts in a receptor-independent manner (panels E-J). Among additional data, we observed that caspase 9 was not activated (Fig 4B) and that tumor cells xenografted *in vivo* also undergo apoptosis under treatment with prazosin (Fig 4D). In addition, we explored whether GICs having escaped a first prazosin treatment are responsive to a second prazosin treatment. The results show that the cells remain sensitive to prazosin treatment albeit at higher concentrations (Expanded View Fig 2D).

We also added in the Expanded View Fig 2 FACS analysis of prazosin-induced GIC apoptosis *in vitro* showing that CD15+ GICs undergo apoptosis (panel A), and TUNEL staining showing increased numbers of cells undergoing apoptosis following *in vivo* prazosin treatment of mice bearing tumors initiated by GBM44 grafting (panel B).

Regarding the relationship between cell cycle inhibition and cell death, immunoblotting of cell cycle proteins following prazosin treatment *in vitro* (GL261 cells) showed no change in the expression of the Cyclin D1, Cyclin D2 and CDK2 proteins, which are required for the G1/S transition. This result suggests that reduced cell cycling of the cells surviving prazosin results from the traumatism induced by prazosin rather than from an organized response of the cell cycle machinery that would precede cell death. This result is now presented in Expanded View Fig 3 that gathers all experiments related to cell proliferation, including the previous panel C of Fig 1 of the original submission (showing decreased BrdU incorporation in GIC treated for 24h with prazosin *in vitro*).

The text was modified accordingly page 5 lines 7-10 as follows:

“Cell cycle was mostly not affected by prazosin. Although we observed a dose-dependent reduction of BrdU incorporation *in vitro* in GICs that had survived to a 24h prazosin exposure, and a decrease in Ki67 staining in tumor grafts of prazosin-treated mice (Fig EV3A-B), no change was observed in cyclin D1, cyclin D3 and CDK2 levels, which are required for G1/S transition (Fig EV3C).”

Referee #3, point 2: “2. The key experiments are based on rottlerin, whose role as a specific PKC inhibitor is highly questionable. It inhibits various kinases including GSK3 β , and uncouples mitochondrial oxidative phosphorylation. To claim the central role of PKC in prazosin-induced apoptosis, additional and more specific inhibitors (e.g. siRNAs) should be explored.”

We addressed this point on Page 3 of this letter together with the answer to referee # 1 point 4, and referee #2 point 7.

Referee #3, point 3: “3. Figs. G-J demonstrate that PRZ treatment results practically in the disappearance of tumors, however, why is survival not substantially prolonged in this case? Are these figures representative? More detailed immunohistological analysis of tumor sections (e.g. for Ki67, TUNEL) would be informative.”

The immunohistological analysis of tumor sections, now Fig 2D and Fig 3D, were done on mice sacrificed at the end of the treatment, and not when mice become morbid (please see the schematic representation of the protocol timings Fig 2A and 3B and the corresponding legend of the figures). We also performed additional TUNEL staining (Expanded View Fig 2B) illustrating a significant increase of apoptosis in prazosin-treated mice.

Referee #3, point 4: “4. It is not stated how many animals were included in the in vivo experiments. From Fig. 2B-E, it seems that there were only 4 mice per group; if this is the case, this number should be statistically justified. Further, the Kaplan Meyer survival for one xenograft model is shown only. What were the results from the second model studied?”

We illustrated pictures of the luminescence on live mice for only part of the individuals used in each experiment. In graphs showing the fold change in total flux, all individual value are presented (8 dots on each graph).

To clarify this point, we have modified the Materials and Methods section as follows (page 12, lines 18-23): “At least 2 mice were euthanized at the end of the treatment for further histological examination. The remaining mice were used to assay survival (at least n=8 per group of treatment). For histological analysis, the brains were kept in 4% paraformaldehyde at 4°C for 24 h, followed by 70% ethanol at room temperature for 24 h. Brains were then embedded in paraffin for 3 h at 67°C. Coronal sections (5 μ m thick) were stained with hematoxylin and eosin and images were acquired (Eclipse E800, Nikon, USA).”

We now present in Fig 2B and Fig 2C the complete data including the fold change in total flux and Kaplan-Meyer curves for GBM005 as well as GBM44.

Referee #3, point 5: “5. Prazosin reduced GICs viability with an EC50 value of 7.88 μ M, several orders of magnitude above the nanomolar concentrations at which it acts on α -ARs. The dose of 5 mg/kg administered to tumor-bearing mice is also significantly higher than the FDA approved. It would be important to test if lower doses, in a range approved by the FDA, will effectively inhibit tumor growth.”

First, we sincerely apologize for the typing error, which led to mention an erroneous dose of 5mg/Kg instead of the 1.5 mg/Kg really used. Of note, in our study, prazosin is not working through adrenergic receptor (not expressed on these cells, see Fig 4G) but through an off-target effect, thus the affinity of the nanomolar range for AR might be not pertinent for the present targeting. We have now tested also a lower dose of prazosin, 0.15 mg/kg, well between the range of the FDA approve regimen (20mg/day). This lower dose resulting in a still significant decrease in tumor growth (Fig 2G), we believe that this result further strengthens the grounds for a clinical use of prazosin as an adjuvant to chemotherapy. These data are now presented on page 4, lines 22-25 and discussed on page 8, lines 15-23. Please see also our response to Referee #1, Major point 4 at the end of page 3 and the beginning of page 4 of this letter.

Referee #3, Minor comments: “The figure legends are not always sufficiently explanatory. For example, what's shown in Fig. 2 G/J? Are those two different xenograft models or two animals per group representing the same model? Fig 2F: it should be labeled when the treatment was initiated.”

We rewrote all the legends and hope to be now as precise and accurate as possible.

CONFIDENTIAL PATIENT INFORMATION DELETED

Thank you for the submission of your revised manuscript to EMBO Molecular Medicine. We have now Thank you for the submission of your revised manuscript to EMBO Molecular Medicine and apologies for the unusual delay in replying, due to difficulties in obtaining the evaluation from on reviewer.

We have now received the enclosed reports from the referees that were asked to re-assess it.

As you will see the reviewers are now globally supportive and I am pleased to inform you that we will be able to accept your manuscript pending the following final amendments:

1) Although we will not be requiring further experimentation at this point, please carefully deal with the remaining comments from reviewers # 1 and 2, with which we agree. Please also carefully check your manuscript for errors. Provided you satisfactorily address these remaining concerns, the final decision will be made at the editorial level. Upon submission, please provide an additional manuscript file in which the amendments are clearly highlighted.

2) As per our Author Guidelines, the description of all reported data that includes statistical testing must state the name of the statistical test used to generate error bars and P values, the number (n) of independent experiments underlying each data point (not replicate measures of one sample), and the actual P value for each test (not merely 'significant' or ' $P < 0.05$ '). I note that you have provided some, but not all P values.

3) We are now encouraging the publication of source data, particularly for electrophoretic gels and blots, with the aim of making primary data more accessible and transparent to the reader. Would you be willing to provide a PDF file per figure that contains the original, uncropped and unprocessed scans of all or at least the key gels used in the manuscript? The PDF files should be labeled with the appropriate figure/panel number, and should have molecular weight markers; further annotation may be useful but is not essential. The PDF files will be published online with the article as supplementary "Source Data" files. If you have any questions regarding this just contact me.

4) Every published paper now includes a 'Synopsis' to further enhance discoverability. Synopses are displayed on the journal webpage and are freely accessible to all readers. They include a short standfirst as well as 2-5 one-sentence bullet points that summarise the paper. Please provide the synopsis including the short list of bullet points that summarise the key NEW findings. The bullet points should be designed to be complementary to the abstract - i.e. not repeat the same text. We encourage inclusion of key acronyms and quantitative information. Please use the passive voice. Please attach this information in a separate file or send them by email, we will incorporate it accordingly. You are also welcome to suggest a striking image or visual abstract to illustrate your article. If you do please provide a jpeg file 550 px-wide x 400-px high.

5) Please note that we now mandate that all corresponding authors list an ORCID digital identifier. You may do so directly through our web platform upon submission and the procedure takes <90 seconds to complete. We encourage all authors to supply an ORCID identifier, which will be linked to their name for unambiguous name identification.

6) I note that the quality of some images especially of the blots is not ideal. In some instances the resolution appears low and the bands appear blocky/blurry when magnifying, in other cases, contrast is excessive and must be decreased (e.g. EV figure 3C and 4B, and others). Please provide better images.

7) Although we have asked you previously, you have not provided the manuscript as a word .doc file. Please comply with this request when submitting your next, final version of the manuscript.

8) Please upload the supplementary figures for Expanded View as separate files.

Please submit your revised manuscript within two weeks. I look forward to seeing a revised form of your manuscript as soon as possible.

***** Reviewer's comments *****

Referee #1 (Remarks):

The authors have largely addressed my major concerns and contributed many new experiments to solve issues raised during the first round of the reviewing process. Hence, my opinion is that the study is now significantly improved. However, two concerns remain:

(1) Characterisation of GBM cells as glioma stem cells (GSCs) is still insufficient. Performing an experiment as shown in Fig. 2A with CD133-high versus CD133-low cells and showing that CD133-high cells have increased potential for tumorigenicity would have proven the case. (2) Prazosin mediates therapeutic effects PARTLY via PKC-delta (as shown in Figs. 5F and G) but there seem to be also other (additional) pathways. Both issues can be addressed without further new experimentation by amending the text: GBM cells are "treatment resistant tumours" (which is a clinically highly important subset of GBM) and the interpretation of PKC-delta as THE mediator of Prazosin should be tamed.

Referee #2 (Remarks):

The authors provide a substantially revised manuscript. The manuscript is interesting and the data presented are strong. I appreciate the efforts that the authors have made to address the concerns raised on the original review. I recognize the challenges that the authors have in addressing some of the points raised. While there are a number of unresolved issues, I believe that the manuscript warrants strong consideration for publication after minor revision.

Remaining concerns:

- The use of multiple models strengthens the general conclusions derived from the manuscript. It is somewhat concerning that there is a strong reliance on a rare variant of glioblastoma. While I agree that the authors have provided additional models, I have some concern about these models. The authors might want to consider using available *in silico* data from patient cohorts to address the relative expression levels and survival patterns for PKCdelta so assure the reader that the effects of PRZ will be likely more general.
- The mechanism remains less than definitive. The studies support the role of PKC, but they did not rule out other mechanisms. I would suggest that they exercise some caution in the claims.
- The authors have provided significant support for their claims, but there remain some issues that are less than definitive. The results suggest that PRZ is effective against all cancer cells, not just initiating cells. This is acceptable, but suggests that the focus may not be ideal in the text. They may want to address this better. Neither PKC, nor PRZ is linked directly to a stem cell program in this manuscript.
- The last two figures are better but still not terribly strong. They may want to include something about patients to improve the impact.

Minor points: There are widespread errors in the text. I would suggest using only the term GIC (GSC is used in the figures but the studies have not been done to address a GSC).

Overall, I believe this study adds to the literature.

Referee #3 (Remarks):

The manuscript has been substantially revised and improved. My concerns have been addressed.

Answers to Reviewer's comments*Referee #1 (Remarks):*

The authors have largely addressed my major concerns and contributed many new experiments to solve issues raised during the first round of the reviewing process. Hence, my opinion is that the study is now significantly improved. However, two concerns remain:

(1) Characterisation of GBM cells as glioma stem cells (GSCs) is still insufficient. Performing an experiment as shown in Fig. 2A with CD133-high versus CD133-low cells and showing that CD133-high cells have increased potential for tumorigenicity would have proven the case.

(2) Prazosin mediates therapeutic effects PARTLY via PKC-delta (as shown in Figs. 5F and G) but there seem to be also other (additional) pathways.

Both issues can be addressed without further new experimentation by amending the text: GBM cells are "treatment resistant tumours" (which is a clinically highly important subset of GBM) and the interpretation of PKC-delta as THE mediator of Prazosin should be tamed.

And Referee #2:

The mechanism remains less than definitive. The studies support the role of PKC, but they did not rule out other mechanisms. I would suggest that they exercise some caution in the claims. The authors have provided significant support for their claims, but there remain some issues that are less than definitive. The results suggest that PRZ is effective against all cancer cells, not just initiating cells. This is acceptable, but suggests that the focus may not be ideal in the text. They may want to address this better. Neither PKC, nor PRZ is linked directly to a stem cell program in this manuscript.

We now modified the text to fulfill the reviewer's requests:

- 1- Abstract page 2 line 5 "Prazosin triggered apoptosis of glioblastoma initiating cells and of their differentiated progeny, inhibited glioblastoma growth in...."
- 2- Results page 3 line 15-16 "A major feature of these cells is their resistance to the currently used chemotherapy temozolomide (Patru et al, 2010)."
- 3- Results page 5 line 11 before the end: "PKCd is involved in Prazosin-Induced GIC Apoptosis"
- 4- Discussion page 6 line 3 before the end "We demonstrate that prazosin-induced GIC apoptosis involves a PKCd-dependent inhibition of AKT pathway."
- 5- Discussion page 7 line 17-18 "We describe here a novel mechanism where prazosin-induced GIC apoptosis includes a mechanism dependent on PKCd activation.."
- 6- Discussion page 7 line 26 "...which may occur in response to PKCd activation.."
- 7- Discussion page 7 line 29 "Moreover, prazosin-induced GIC apoptosis is mostly dependent on PKCd activation.."
- 8- Discussion page 8 line 7-8: "activation of PKCd is mandatory for prazosin to exert its pro-apoptotic action. The possibility that additional molecular mechanism are involved in prazosin-induced cell death cannot be excluded but remains to be elucidated."

Referee #2 (Remarks):

The authors provide a substantially revised manuscript. The manuscript is interesting and the data presented are strong. I appreciate the efforts that the authors have made to address the concerns raised on the original review. I recognize the challenges that the authors have in addressing some of the points raised. While there are a number of unresolved issues, I believe that the manuscript warrants strong consideration for publication after minor revision.

Remaining concerns: The use of multiple models strengthens the general conclusions derived from the manuscript. It is somewhat concerning that there is a strong reliance on a rare variant of glioblastoma. While I agree that the authors have provided additional models, I have some concern about these models. The authors might want to consider using available in silico data from patient cohorts to address the relative expression levels and survival patterns for PKCdelta so assure the reader that the effects of PRZ will be likely more general. The last two figures are better but still not terribly strong. They may want to include something about patients to improve the impact.

We thank the reviewer for this suggestion and analyzed the TCGA transcriptome dataset of primary glioblastoma, and found that high levels of PKCdelta are correlated with shortened overall survival and progression-free survival. This result is described now in the text as follows, and shown in Fig. EV5.

Results section page line 20-23: “Interestingly, analysis of mRNA profiles of adult glioblastoma available in the TCGA dataset showed that high expression of PKCd is associated with a poorer prognosis for patients. High PKCd (*PRKCD*) mRNA levels were inversely correlated with overall survival as well as progression-free survival, (Fig EV5).”

Legend for Fig EV5 (page 22): “**Expanded View Figure 5 - PKCd expression is associated with a poorer prognosis in human patients.** Analysis of the TCGA dataset revealed that *PRKCD* transcript levels are inversely correlated with the overall (A) and progression free (B) survival of adult glioblastoma patients (the analysis was restricted to the samples of untreated patients, logrank test, TCGA cohort, pvalue).”

Minor points:

There are widespread errors in the text.

We carefully checked the manuscript and hope the misspellings were all corrected

I would suggest using only the term GIC (GSC is used in the figures but the studies have not been done to address a GSC).

We corrected figure 1

YOU MUST COMPLETE ALL CELLS WITH A PINK BACKGROUND ↓**USEFUL LINKS FOR COMPLETING THIS FORM**

Corresponding Author Name: Hervé Chneiweiss and Marie-Pierre Junier

Manuscript Number: EMM-2015-05421